# Long-term Lidar Observations of the Gravity Wave Activity near the Mesopause at Arecibo

Xianchang Yue[1,2], Jonathan S. Friedman[4], Qihou Zhou[5], Xiongbin Wu[1,2], Jens Lautenbach[3]

[1]School of Electronic Information, Wuhan University, Wuhan, 430072, China
[2]Collaborative Innovation Center of Geospatial Technology, 129 Luoyu Road, Wuhan, 430072, China
[3]Arecibo Observatory – University of Central Florida, Arecibo, Puerto Rico.
[4]Puerto Rico Photonics Institute, School of Science and Technology, Universidad Metropolitana, Cupey, Puerto Rico.
[5]Electrical and Computer Engineering Department, Miami University, Oxford, Ohio, USA

*Correspondence to*: Xianchang Yue (yuexc@whu.edu.cn)

**Abstract.** 11-years long K Doppler lidar observations of temperature profiles in the mesosphere and lower thermosphere (MLT) between 85 and 100 km, conducted at the Arecibo Observatory, Puerto Rico (18.35°N, 66.75°W), are used to estimate seasonal variations of the mean temperature, the squared Brunt-Väisälä frequency, $N^2$, and the gravity wave potential energy in a composite year. The following unique features are obtained: 1. The mean temperature structure shows similar characteristics as a prior report based on a smaller dataset: 2. Temperature Inversion Layers (TILs) occur at 94-96 km in spring, at ~ 92 km in summer and at ~ 91 km in early autumn; 3. The first complete range-resolved climatology of Gravity Wave Potential Energy (GWPE) derived from temperature data in the tropical MLT exhibits an altitude dependent combination of annual oscillation (AO) and semiannual oscillation (SAO). The maximum occurs in spring and the minimum in summer, a second maximum is in autumn and a second minimum in winter; 4. The GWPE per unit volume reduces below ~ 97 km altitude in all seasons. The reduction of GWPE is significant at and below the TILs but becomes faint above, this provides strong support to the mechanism that the formation of upper mesospheric TIL is due mainly to the reduction of GWPE. The climatology of GWPE shows an indeed pronounced altitudinal and temporal correlation with the wind field in the tropical mesopause region published in the literatures. This suggests the GW activity in the tropical mesopause region should be manifested mainly by the filtering effect of critical level of the local background wind and the energy conversion due to local dynamical instability.

## 1 Introduction

Gravity Waves (GWs) are believed to be the primary force driving the large-scale circulation and coupling of different atmospheric layers due to their momentum and energy deposition when breaking or dissipating (Fritts and Alexander, 2003; Lindzen, 1981; Smith, 2012). GWs often originate from copious tropospheric sources and propagate upward. Their amplitudes grow rapidly with altitude under the condition of decreasing atmospheric density. GWs begin to break or dissipate in the Mesosphere and Lower Thermosphere (MLT) where their influences have been shown to be strong by

various observations (e.g., Baumgarten et al. 2015; 2017; 2018; Cai et al., 2014; Gardner and Liu, 2010; Li et al., 2010; Lu et al., 2009; Yuan et al., 2016). GWs are therefore an essential component in current General Circulation Models (GCMs) [e.g., Liu and Meriwether, 2004; Picone et al., 2002]. Associated modeling studies have shown that including the effects of GWs is a key point to simulate realistic quasi-biennial and semiannual oscillations and other phenomena in the stratosphere and

mesosphere [e.g., Dunkerton, 1982; Huang et al. 2010; Xu et al., 2006].

The upward propagations of GWs are often influenced by the background wind fields (e.g., Yiğit and Medvedev, 2015). The altitude ranges, where GWs interact with the background winds to dissipate and deposit energy and momentum to the background, are of high interest to researchers. This happens mainly in the middle atmosphere near or between stratopause and mesopause. Above 35 km altitude, the semiannual oscillation (SAO) shows a predominance in the annual variability of

the zonal winds (e.g., Garcia et al., 1997; Hirota, 1980). The SAO phase of zonal winds shifts approximately 180° with the altitude increasing from stratosphere to mesosphere. The stratospheric SAO leads to a seasonal variation of filtering of the upward propagating waves, which results in a specific seasonal variation of GW activity in the mesosphere.

The mean zonal winds in mesosphere and lower thermosphere (MLT) have been measured by both ground-based radars (e.g., Garcia et al., 1997; Lieberman et al., 1993) and satellite-based instruments such as the High-Resolution Doppler

Imager (HRDI), Wind Imaging Interferometer (WINDII) and the Microwave Limb Sounder (MLS) on board the Upper Atmosphere Research Satellite (UARS) or the Doppler Interferometer (TIDI) on board the Thermosphere Ionosphere Mesosphere Energetics and Dynamics (TIMED) satellite (e.g., Smith, 2012). They show different features of the zonal winds in the mesopause range from tropical region to middle latitude regions. Garcia et al. (1997) and Smith (2012) showed that, for example, the westerly winds prevailed in the range 80-95 km both in January and July in the HRDI equatorial zonal wind

but it reversed below or above this range in the north hemisphere. The monthly mean HRDI equatorial zonal wind showed that, the easterly winds were prevailing in equinoxes seasons in the 80-100 km altitude range with the minimal wind speed occurring near 92 km, while the westerly winds prevailed in solstice seasons in the 80-94 km altitude range with the maximal wind speed occurring near 88 km. The easterly winds turn to be prevailed above 95 km in solstice seasons (Smith, 2012). Therefore, the zonal winds are low or zero around 92 km altitude in tropical region. The zero-wind lines will enhance

reducing or dissipating of zonal propagating gravity wave with low to moderate phase speed. There are also reports about the tropical MLT mean winds measured by ground base radars (e.g., Guharay and Franke, 2011; Li, et al., 2012).

Temperature is a crucial parameter indicating the state of the atmosphere. To measure the temperature in the MLT region, satellite and lidar techniques have been developed in recent decades. Satellite measurements have the advantage of resolving large spatial scale wave structures, but the short-term variability in dynamical features gets lost. While for the lidar

measurements, they provide vertical profiles of temperature with a suitable temporal and vertical resolution at a particular location. Lidar data can well resolve gravity waves with short and medium periods and their temporal development. The perturbation or standard deviation from zonal mean temperature is often used as a wave activity indicator in the atmosphere. For instance, Offermann et al. (2006) use the temperature measured from CRISTA (CRyogenic Infrared Spectrometers and Telescopes for the Atmosphere) and from SABER (Sounding of the Atmosphere using Broadband Emission Radiometry) to

investigate the global wave activity from upper stratosphere to lower thermosphere. They showed quite different wave behaviours below and above their defined 'wave-turbopause' close to but lower than the mesopause. Below the turbopause, the propagation of the wave is significantly dissipated, while above that, the propagation of the wave is almost free.

The reduction of GW activity has also been presented by lidar temperature measurements at mid-latitude sites (Mzé et al., 2014; Rauthe et al., 2008). Both Mzé et al. (2014) and Rauthe et al. (2008) showed that the GW activities presented an annual variation with maximum in winter and minimum in summer, they reduced significantly at the upper mesosphere altitudes above ~ 70 km in all seasons, while the reduction below ~ 70 km is not so evident. Mzé et al. (2014) observed a nearly undamped propagation of GW in summer in the low mesosphere. Meanwhile, Rauthe et al. (2008) also reported the weakest reduction occurring in the summer seasons. Since the effects of GW in the numerical climate and weather prediction models are usually represented simply by parameterization (Kim et al., 2003). there are still large discrepancies between model and measurement results (Geller et al., 2013). Therefore, more attention should be paid to the GW parameterization about these kinds of observations in the upper mesosphere and mesopause region to improve the model results.

Seasonal variations of GW activities based on lidar temperature measurements have been investigated at a few low latitude sites (Chane-Ming et al., 2000; Li et al., 2010; Sivakumar et al., 2006). These studies used Rayleigh lidar data and focused mainly on the upper stratosphere and lower mesosphere in an altitude region from 30 to 80 km. They showed the SAO dominated the seasonal variability of GW activities with maxima in both winter and summer and minima near the equinox. Li et al. (2010) related this to the dominated SAO in the mean zonal wind in the tropical stratosphere and lower mesosphere. Considering the different transition of the mean zonal wind in the range 80-95 km in the tropical region noted above, the seasonal variability of GW activity in the tropical upper mesosphere is expected to be different from either the lower altitudes or the mid- and high-latitude regions.

To our knowledge, the seasonal variability of GW activity retrieved from mesospheric lidar temperature data with high temporal and vertical resolution have never been reported from a tropical location.

Such measurements of atmospheric temperature have been conducted with the K Doppler lidar located at the Arecibo Observatory (18.35°N, 66.75°W), Puerto Rico (Friedman and Chu, 2007; Yue et al., 2013, 2016,). Since December 2003, the Arecibo Observatory K Doppler lidar has operated routinely, producing high-quality temperature data. In this report, we estimate the mean temperature, the squared Brunt-Väisälä frequency, $N^2$, and the GW potential energy and their annual variability from the temperature dataset. We find an altitude-to-altitude close relationship between the annual variability of the potential energy in this report and that of the mean zonal winds in the literature. Their implications to the GW activity in the MLT are discussed.

## 2 Observations

The Arecibo K Doppler lidar probes the potassium D1 line to deduce the potassium density and neutral air temperature simultaneously by employing the three-frequency technique (e.g., Friedman and Chu, 2007; Friedman et al., 2003). The

temperature was obtained at 0.45/0.9 km vertical and 10/30 minutes temporal resolution, respectively (Friedman and Chu, 2007), and we unify them into profiles with resolutions of 0.9 km and 30 minutes in vertical and temporal dimensions, respectively. This data processing excludes the perturbations relevant to GWs with vertical wavelengths and observed periods less than 1.8 km and 1 hour, respectively. Their exclusion may bias the deduced GW associated potential energy estimations towards mid- and low-frequency GWs. The root mean square (RMS) temperature errors is about 2 to 3 K at the peak of the K density layer and increase to about 10 K around 85 and 100 km (Friedman and Chu, 2007; Yue et al., 2017). Measurements were made only at night in two periods, one from December 2003 to January 2010, the other from November 2015 to April 2017. Data were available for every calendar month. Across the 11 years of the data collection period, 200 observing nights with a total of 1470 hours of data were available at the time of preparing this report. The number of observation nights/hours varies from 4 nights/28 hours in October, to 32 nights/259 hours in January. On average there are 16.7 nights or 122.5 hours of observation each month. The statistics for the used temperature observations are plotted in Fig. 1 and the numbers of observational nights and durations are summarized for each month in Table 1. When the data are binned into weekly intervals, they cover 45 weeks of a year. On average there are 4.4 nights or 32.7 hours of observation each week. The gaps are at weeks 7,18, 19, 25, 26, 39 and 42. The distribution of the data is quite even during the year. This allows us to fit the GW activity annual variability through weekly means of temperature.

## 3 Calculating Methods

The potential energy density $E_p$ can be estimated from temperature observations and then is chosen as a measure of GW activity. $E_p$ is defined as (see, e.g., Vincent et al., 1997)

$$E_p = \frac{1}{2}\left(\frac{g}{N}\right)^2 \left(\frac{T'}{\overline{T}}\right)^2,$$  (1)

where

$$N^2 = \frac{g}{\overline{T}}\left(\frac{d\overline{T}}{dz} + \frac{g}{C_p}\right),$$  (2)

Here $g$ equals to 9.5 ms$^{-2}$, is the gravitational acceleration in MLT; $N$ is the Brunt-Väisälä frequency calculated according to Eq. (2); $\overline{T}$ is the mean temperature averaging over altitude; $T'$ is the temperature perturbation; and $C_p$ is the constant-pressure heat capacity, equals to 1004 JK$^{-1}$kg$^{-1}$. In Eq. (1), the calculation of $E_p$ depends on the estimations of $N$, $\overline{T}$ and $T'$. The procedure adopted by Gardner and Liu (2007) is closely followed here for the estimation of $T'$. It includes 4 steps. Step 1: for the unified data with 0.9 km and 30 minutes in each night of observation, data points with photon noise errors larger than 10 K in temperature are discarded. Step 2: the linear trend in time at each altitude in the temperature profiles is then subtracted to eliminate the potential biases associated with GWs with periods longer than about twice the observation period. Step 3: deviations exceeding three standard deviations from the nightly mean are discarded from the resulted temperature

deviation series at each altitude to remove occasional outliers. Step 4: the vertical mean is subtracted from each temperature deviation profile to eliminate the influences of the waves with vertical wave length longer than about twice the profile height range (~ 25 km). The resulted temperature deviations ($\Delta T$) are used to calculate nightly mean temperature perturbation ($T' = \overline{|\Delta T|}$).

The data on 18 December 2003 are taken as an example of the data processing. Fig. 2a shows the temperature contours after step1. The data covers largely 82-103 km altitude range, the bottom edge increases to about 84 km in the second half of the night. Fig. 2b show the subtracted trends in time which indicates that the background temperature decreases at each altitude between 85-100 km range but increases near both the bottom and top edges. Fig. 2c is the corresponding temperature deviations. The temperature deviations after step 4 are shown in Fig. 2d. It shows that, in contrast to mid- and high latitudes

(e.g. Baumgarten et al., 2018; Rauthe et al., 2008), the perturbations (GWs) are dominated by small-scale variability and to a smaller extend by coherent wave structures. The difference between Fig. 2c and Fig. 2d is pronounced. This indicates that step 4 is effective in eliminating the influences of the waves with longer vertical wave length. Furthermore, Guharay and Franke (2011) have given a rather strong semidiurnal tidal amplitude at the mesopause region through the meteor radar observations over a nearby site (20.7°N, 156.4°W). The amplitude increases with altitude and shows a clear SAO pattern

with maxima during solstices. This kind of feature is not found in the variations of GW potential energy with altitude and season in later sections here. However, the total effective of this method in removing the tidal is not easily to evaluated due to the lack of the knowledge on tidal components in this latitude and the shorter and often intermitted measurement periods.

Fig. 3a shows the individual profiles of the temperature fluctuations (cyan curves) for the night. The nightly mean temperature perturbations and the statistical uncertainties are also presented by black and red lines, respectively. The

20 statistical uncertainties are usually less than 5 K below 95 km altitude, they increase a bit to ~ 6 K and keep it above. While the nightly mean temperature perturbations decrease from ~18 K at 85 km to ~6K at 89.5 km and then increase to and oscillate near 11 K above. The square of the ratio of temperature statistical uncertainty over nightly mean perturbation is plotted in Fig. 3b. It represents the relative error in estimating the potential energy due to statistical uncertainty in the temperature data. In this night, the relative errors are lower than 20% below 95 km and become larger above. The maximal

relative error occurs just below 97 km with a value of 59%. The relative error is a bit larger than 30% near 100 km altitude but less than 10% near 85 km.

The unified 0.9 km and 30 minutes resolution temperature profiles on each observational night in the same folded week are binned and averaged to the fixed vertical and temporal grids to construct the weekly composite night of temperature of a year. The weekly composite night data of $\overline{T}$ is first spatially and then temporally smoothed using Hamming windows with full

widths at half maximum (FWHM) of 2.7 km and 3 hours, respectively. This method of computing the weekly composite night follows the approach used by Friedman and Chu (2007) and the references therein. The resulting weekly composite night of temperature usually covers most of the night from sunset to sunrise (not shown here, the resulting monthly composite night can be seen in Figure 3 of Friedman and Chu (2007), which is computed with a small part of the data set used here). It is a close representation of the mean state at the fixed time and altitude bins within an averaging week night.

After that, the weekly composite nights are then averaged to derive the weekly mean profiles. The nightly mean temperature perturbation profiles with unified 0.9 km resolution in the same folded week are averaged to obtain the weekly mean temperature perturbation profile; The weekly mean profiles of $N^2$ and the consequential $E_p$ at 0.9 km resolutions are estimated through Eq. (2) and (1), respectively. For the weekly mean profiles of each parameter, they are fitted to a harmonic fit model including the annual mean plus 12-month (annual) oscillation and 6-month (semiannual) oscillation. The equation of the model is as following:

$$\Psi(z,t) = \Psi_0(z) + A_{12}(z)\cos\left[\frac{2\pi}{365/7}\left(t - \varphi_{12}(z)\right)\right] + A_6(z)\cos\left[\frac{4\pi}{365/7}\left(t - \varphi_6(z)\right)\right] \qquad (3)$$

where $\Psi(z,t)$ is the value of a weekly mean parameter at altitude $z$ and week $t$, expressed in week of the year (1-52), $\Psi_0(z)$ is the annual mean, $A_n(z)$ and $\varphi_n(z)$ ($n = 6,12$) are the amplitude and phase of the $n$-month oscillation, respectively.

To keep the characteristics of higher order components in the seasonal climatology of each parameter, the fitted $\Psi(z,t)$ was subtracted from the raw weekly mean profiles and the residuals were smoothed using a Hamming window with a FWHM of 4 weeks. The smoothed residuals were then added back to the fitted $\Psi(z,t)$, the resulting seasonal climatology (hereafter SC) of each parameter is then illustrated together with weekly mean profiles in the results section. Meanwhile, the fitted seasonal variations (hereafter SV) $\Psi(z,t)$ of each parameter, i.e. the mean plus the annual and semiannual harmonic fits, are also presented together with the amplitude and phase information of each monochromatic wave in the following section.

## 4 Results

In the following the results of the analysis are shown and discussed with respect to the seasonal variation of the mean temperature (section 4.1), Square of the Brunt-Väisälä frequency (section 4.2) and the GW activity (section 4.3). Therefore, we plot the weekly mean profiles and the corresponding SCs of $\overline{T}$, $N^2$, and $E_p$ in Fig. 4, 6, and 8, respectively, at 0.9 km vertical and one-week temporal intervals. The SVs of $\overline{T}$, $N^2$, and $E_p$ by applying Eq. (2) are presented in Fig. 5a, 7a, and 9a (top left panel in each figure), respectively. The amplitudes and phases of the 12-month and 6-month oscillations of the regression model are plotted in Fig. 5c, 7c and 9c (top right panels) and 5d, 7d and 9d (bottom right panels), respectively. In the raw data, temperature error due to photon noise is usually less than 5 K in the altitude range 87-97 km because the K density in this range is usually rather larger (e.g., Yue et al., 2017). To show the seasonal variation of each parameter more clearly, the data between 87 and 97 km are averaged in altitude and then fitted to the same seasonal model consisting of the annual mean, AO and SAO. Note that it is in a risk of smoothing the temporal variations for those parameters, such as $N^2$, which is strongly varying in term of altitude in this range. The averaged results and the fitted curves are plotted in Fig. 5b, 7b and 9b (bottom left panels). The statistical parameters for these fits are summarized in Table 2.

## 4.1 Seasonal Variation of the Mean Temperature

The weekly mean temperature $\overline{T}$ illustrated in Fig. 4a and the SC of temperature illustrated in Fig. 4b shows that the Arecibo climatology is warmer in late autumn and early winter and colder in summer throughout all altitudes. Fig. 5a shows the seasonal variation of temperature synoptically. Temperature Inversion Layers (TILs) present in most of the time. The altitudes where the temperature gradient changes from positive to negative are represented by the black cross in both Figures 4b and 5a. The deviations of TILs between these two figures are minor for most of the time. Significant differences occur mainly in the period between September and December. The inversion layers near 92 km in late September and near 89 km from late November to early December in Fig. 4b vanish in Fig. 5a, meanwhile, the inversion layer near 94 km during the first half of September changes to ~ 92 km in Fig. 5a. It is noted that the temperatures vary slightly from 85 to 94 km altitude in most time from September to December. The inversion layers evaluated by finding the zero value of temperature gradient in this period are not confident enough in case of small perturbation existed, therefore, we ignore this period from September to December when discussing about the TILs. However, there is a common feature that temperature maximum occurs at ~ 97 in October and November in both figures 4b and 5a. To simplify the analysis on the influences of different harmonic components, we take Fig. 5a as the example in the following descriptions and discussions. In spring, TILs occur at ~96 km from late February to April. In summer, TILs appear at ~ 91 km from late May to the first half of September. In winter, TILs occur at ~94 km from January to the first half of February. The minimal temperature occurs near 98.5 km at most time except for the period from the second half of September to November where it is situated at ~ 96 km. This should be the mesopause according to the result that the mesopause is at the 95-100 km level at low-latitude obtained by SABER observations (Xu et al., 2007a).

Figure 5c shows that the amplitudes of the AO are obviously larger than that of the SAO. Fig. 5d shows that the phase (defined as the time of the maximum perturbation) of AO oscillates in a range between day of year (DOY) -60 and 10, while that of SAO varies between DOY 110 and 150. Fig. 5b shows that the mean temperature is warmest between October and November and coldest in July. A secondary peak/trough occurs in April/February.

Notice that the warmest temperature occurs around October with shortest observation times which reduce the confidence level of the harmonic fit. However, the observation times in both September and November are longer than 100 hours in more than 10 nights, they help to keep the confidence level of the harmonic fit. Moreover, the temperature structure shown in Fig. 5a agrees well with the temporal variations of the equatorial zonal mean temperature in the range 85-100 km observed by SABER (Xu et al., 2007b). The amplitudes and phases of both SAO and AO observed by SABER at 20° N latitude had been shown by Xu et al. (2007b) in their middle panels of Figure 10. Comparisons show that the lidar observed phases of both SAO and AO shown in Fig. 5d agree with those obtained by SABER in the same altitude range. The SAO amplitude shown in Fig. 5c agrees quite well with that observed by SABER in both magnitude and vertical structure. The lidar AO amplitude shows similar vertical structure with that of SABER, but the magnitude of lidar AO amplitude is at least

1 K larger than that observed by SABER. The agreement between lidar and SABER observations gives us more confidence to use the lidar observed temperature data studying the GW activities in latter sections.

Except for the smaller amplitude of SAO oscillation in this study, the phase of SAO, the amplitude and phase of AO of the mean temperature is consistent with a previous study by Friedman and Chu (2007) (see their Fig. 6 to 7), who used data collected between December 2003 and September 2006. Comparing Fig. 4b here to the Fig. 6 in Friedman and Chu (2007), there are some different features in these two climatology results. For example, the temperatures are about 10 K warmer in the winter months of December and January in this study. The differences are caused by two reasons. The first reason is the much more extensive data set from year 2003 to 2017 covering a whole solar period here. The second reason is that the harmonic fit model is applied to weekly averages here, while it was applied to monthly averages in Friedman and Chu (2007).

## 4.2 Square of the Brunt-Väisälä frequency

The square of the Brunt-Väisälä frequency $N^2$ is a good indicator to characterize the atmospheric static stability. Gardner and Liu (2007) indicated that the resulting $N^2$ was usually overestimated in this way due to the eliminations of gravity waves when the weekly mean temperature profiles were derived by employing data averaging. However, they pointed out that the lower-value regions of $N^2$ represented well the lower stability of the atmosphere, i.e., the greater wave dissipations. Figures 6 and 7a show that $N^2$ is highly variable with season and altitude. Take the summer months from June to August as example, Fig. 7a shows that $N^2$ is quite low near the bottom and then increase quickly with altitude. It is obviously large between 87 and 92 km altitudes and decreases above. It is small in the 94-98 km range and turn to increase gradually near the top boundary. The seasonal variations are obviously different every 4-km-thick range from 87 to 96 km altitude. The features shown in Fig. 6b are more complicated than Fig. 7a. Assuming an isothermal atmosphere with the background temperature being 190 K, $N^2$ is estimated to be about $4.7 \times 10^{-4}$ s$^{-2}$ which is represented largely by the orange color in figures 6b and 7a. Therefore, below about 96 km, the regions with red color (larger than $4.7 \times 10^{-4}$ s$^{-2}$, indicating a positive temperature gradient with altitude) agree with the TILs as expect. It shows clearly that the TILs occur at about 92-95 km altitude range in February and March while they occur at about 87-92 km altitude range through the summer months. A low value of GW potential energy is expected in the region with $N^2$ being large in case that other parameters keep unchanged.

The fitted curve of $\overline{N^2}$ (average of $N^2$ between 87 and 97 km height) as shown in Fig. 7b exhibits a seasonal variation. The maximum occurs in July and the minimum between October and November, while a secondary maximum occurs between January and February and a secondary minimum in May. However, it is noted that the seasonal variations of $N^2$ vary highly with altitude. This 87-97 km mean fitting curve cannot represent the whole features of $N^2$ in this altitude range.

Figure 7c shows that the amplitudes of the 12-month and the 6-month oscillations are comparable throughout most of the altitude range of interest. They oscillate similarly and look like sinusoids with troughs occurring at altitudes ~87 km, ~ 92km, ~96 km, ~98 km. The phases of these two components illustrated in Fig. 7d also show quick transition at these altitudes.

## 4.3 Seasonal Variation of the Gravity Wave Activity

GW activity is directly manifested by the wave energy. Figure 9a shows the contour plots of the harmonic fitted GW potential energy $E_p$ from lidar observation. $E_p$ is coloured in a logarithmic scale (log10). It can be seen that, below 97 km altitude, $E_p$ always reaches the maximum in equinox seasons, and mostly near spring equinox. More interesting, in equinox seasons the potential energy decreases with altitude from the bottom to ~ 91 km and then shifts to increase with altitude in the range from 91 to 95 km. However, the potential energy is quite smaller in the altitude range 85-95 km during solstices. The energies are low near 91 km throughout all the year. Above 91 km and below 97 km an obvious SAO is visible in Fig. 9a. The oscillations of potential energy become very weak around 97 km. The SC of potential energy illustrated in Fig. 8b shows some difference. The seasonal variations of $E_p$ exhibit an oscillation with period of ~ 3 months at 87-95 altitude range. A dominant maximum occurs from the second half of March to the first half of April. Several minima occur at time January-February, May-June, August-September, and November-December. This implies that some higher order oscillation may also play important role besides the AO and SAO. We note this feature of $E_p$ varying with higher oscillation pattern and leave it for further discussion. We only focus on AO and SAO in this paper.

Figure 9c shows that the amplitudes of both the 12-month and the 6-month oscillations are comparable at most altitudes. The amplitude of SAO becomes smaller in the altitude range 94-98 km. It decreases to almost 0 J.$\text{kg}^{-1}$ at 97-98 km. Fig. 9d shows that the phase of SAO is almost independent of altitude. It is quite close to DOY 100 at most altitude. The AO has almost the same phase as the SAO from 88 to 96 km altitude. Its phase shifts to the end of the year near the top edge where it is the dominant seasonal variation.

In the altitude range 87-97 km, Fig. 9c shows that the amplitudes of both AO and SAO vary slightly, meanwhile, Fig. 9d shows that the phases of both AO and SAO vary also slightly. Therefore, the mean $E_p$ averaged in the altitude range 87-97 km and the corresponding harmonic fit shown in Fig. 9b can represent the seasonal behaviour of $E_p$ in this altitude range well. The harmonic fit curve of $E_p$ shows a combination of AO and SAO with the maximum of 404 J.$\text{kg}^{-1}$ near vernal equinox and the minimum of 264 J.$\text{kg}^{-1}$ in the end of November. The maximum is a factor of 1.5 larger than the minimum.

## 5 Discussion

### 5.1 Mesospheric Temperature Inversion Layer

We have noticed that obvious TILs occur at ~96 km in spring, at ~ 91 km in summer, and at ~ 92 km in early autumn. The TILs in the upper mesosphere over Arecibo had been reported both in case study and/or study of climatology by using subset of the data used in this study (Yue, et al., 2016; Friedman and Chu, 2007). The formation mechanism for TIL in the mesopause region had been reviewed by Meriwether and Gerrard (2004). One primary mechanism for the upper mesosphere TIL is that upward propagating GWs reach a critical level via interaction with the background flow and/or tides. The GW

potential energy accumulates with the wave compressed in reaching to the critical level. Xu et al. (2009) analysed satellite observations and showed that the DW1 tide interacted with GW leading to the damping of both DW1 tide and GW, the larger the amplitude of DW1, the larger the damping. Consequently, the occurrence of TIL and the decrease of the GW $E_p$ are expected at and just below the locations where DW1 amplitude is large. Climatology of WACCM Simulations showed that, at 20° N, both the zonal and meridional components of DW1 tide amplitudes are large in height range 80-100 km around vernal equinox and in altitude range 90-100 km in summer months from June to August (e.g. Fig. 10 in Smith, 2012). These areas with large DW1 tide amplitude in their Fig. 10 match perfectly with the TILs in Fig. 5a.

## 5.2 Reduction of GW Potential Energy

For freely propagating GWs, the potential energy per unit mass ($J.\,kg^{-1}$) should increase exponentially with altitude for the conservation of energy. Fig. 9a shows that the potential energy decreases firstly and then turn to increases gradually with altitude below ~ 97 km in all seasons. Above ~ 97 km, the GW potential energy enhanced significantly with altitude. This behaviour of mean potential energy is much similar to that retrieved from satellite temperature data (Offermann et al., 2006, see their Figures 10 and 11). The altitude of ~ 97 km is in the vicinity of their 'wave-turbopause' altitude range, is close to the mesopause over this site (Friedman and Chu, 2007; Xu et al., 2007a; Yue et al., 2017), and is the level where the seasonal variation of zonal winds changing from clear SAO pattern to AO dominant pattern (e.g. Li et al., 2012). This result suggests that a possible mechanism for the GW energy dissipation, i.e., the GW dissipates or deposes energy or momentum below about the mesopause (or the wave-turbopause defined by Offermann et al. 2006). This conjecture should be taken with caution Because the relative error in the estimated $E_p$ could reach 30% or even larger due to the statistical uncertainty of temperature measurement at this altitude range. But the quantitative influence of this uncertainty on the energy increase with altitude is hard to evaluate because the statistical uncertainty does not show an increase trend at 96-100 km altitude range as shown in Fig. 3b. In addition, it needs point out that the increasing $E_p$ above 97 km in spring, autumn and winter months matches with the increasing easterly winds in this altitude range and seasons as discussed in the 5.3 subsection.

To learn in depth the dissipation of GW in the mesopause region at Arecibo, we multiplied the harmonic fitted $E_p$ with the air density taken from the CIRA-86 reference atmosphere [Fleming et al., 1990], and averaged the weekly profiles every 13 weeks (period of a season) centring at each equinox or solstice. The resulted profiles of the potential energy per unit volume (in $J.\,m^{-3}$) in four seasons are plotted in Fig. 10. If GWs propagate upward without energy dissipation, the lines of energy per unit volume would be vertical. Therefore, the overall left-sloping lines in Fig. 10 indicate that the reductions of GW potential energy occur below ~97 km in all seasons. The reduction of GW potential energy in the mesosphere had been reported by lidar observations at other latitude stations (e. g. Mzé et al., 2014; Rauthe et al., 2008). Both observations of Mzé et al. (2014) and Rauthe et al. (2008) indicated dissipation of GW $E_p$ throughout the mesosphere in all seasons.

The reduction of GW $E_p$ indicates the deposition of GW energy and momentum into the background atmosphere, which would lead to the increase of background temperature and/or even the occurrence of TIL. This drives us to investigate the

relationship between the reduction of GW $E_p$ and the temperature structure in depth. We are excited to find that each profile of the GW potential energy per unit volume ((in $J.m^{-3}$) as shown in Fig. 10 shows a rapider reduction of energy at and below the TIL altitude of the corresponding season and turns to a much slower reduction and/or even conservation of energy above. For examples, the behaviours of the green curve (profile for winter) around 94 km altitude (the altitude of TILs in winter), the blue curve (profile for summer) around 91 km altitude (the altitude of TILs in summer) and the black curve (profile for spring) around 97/98 km altitude (~1 km above the altitude of TILs in spring). These close connections of the mesospheric TILs with the reduction of GW potential energy provide strong support to the mechanism that the upper mesosphere TIL formed due to the interaction of GW with the upper mesospheric wind/diurnal tides through critical level effects. From other point of view, the strong gradient change in the seasonal mean profiles of GW potential energy per unit volume should be induced strongly by the horizontal wind field in this region.

## 5.3 Seasonal variations of GW Potential Energy

We point out a semi-annual cycle of GW $E_p$ with maximum in spring and minimum in summer and a second maximum in autumn and a second minimum in winter in the altitude range 87-97 km. The maximum of the GW $E_p$ alters to autumn below 87 km and above 97 km altitude. These results agree with the observations at other low-latitude stations. Gavrilov et al. (2003) studied the GW seasonal variations by using Medium-Frequency (MF) radar observation over Hawaii (22°N, 160°W). They found a semiannual variation of GW with the maximum intensity at the equinoxes above 83 km, the mean zonal wind had also a mainly semiannual variation in this altitude range. The seasonal variations of GW activities at low-latitude stations are different to those obtained from lidar observations at other latitude stations in the upper mesosphere (Mzé et al., 2014; Rauthe et al., 2006, 2008). Rauthes et al. (2008) provided the seasonal variations of GW $E_p$ at a station of 54°N latitude by using a 6-years of lidar temperature observations from 1 to 105 km. They showed an annual-dominated variation of GW $E_p$ with the maximum in winter and the minimum in summer in the mesopause region. Mzé et al. (2014) reported a semi-annual variation of GW $E_p$ with maxima in winter and in summer and minima during the equinoxes in the upper mesosphere (~75.5 km) by using Rayleigh lidar observations from 1996 to 2012 at a mid-latitude station (~44°N). They showed that the maximum of $E_p$ was about 144 $J.kg^{-1}$ on average at 75.5 km in August while the minimum of $E_p$ is about a factor of 2.5 smaller than the maximum. The ratio between the maximum and the minimum is obviously larger than that of 1.5 in the altitude range 87-97 km at Arecibo.

The cause of the observed seasonal variations of GW activities in the mesosphere was discussed by several authors. One that is often concerned is the influence of critical level filtering of GW by the background wind (Lindzen, 1981, Yue et al., 2005). Gavrilov et al. (2003) had attributed their observed semiannual variation of GW intensity to the dependence of GW generation and propagation on the background wind and temperature by numerical simulations. In a mid-latitude station Juliusruh (55°N,13°E), Hoffmann et al. (2010) reported a semiannual variations of GW activity in the upper mesosphere and lower thermosphere with maxima in winter and summer and minima during equinoxes by using MF radar measured

winds. This seasonal dependence is assumed to be mainly due to the filtering of GW by the background wind in the stratosphere and lower mesosphere. It is not always the case. Rauthe et al. (2008) did not find a direct correlation between the strength of the GW activity and the background wind direction and/or wind speed taken from European Centre for Medium-Range Weather Forecasts analysis.

5    Here we also want to check the relation between our observed GW activity and the wind direction and/or wind speed. Some scientific literatures reported studies about seasonal variation of mean zonal wind in the tropical mesopause region (see e.g., Fig. 3 in Garcia et al. 1997; Fig. 1 in Li et al., 2012; Fig. 3 in Smith 2012). The monthly mean HRDI equatorial zonal wind showed that, the easterly winds were prevailing in equinoxes seasons near 80 km altitude. They then decreased with altitude from 80 km above and turned to increase above ~ 92 km, while the westerly winds prevailed in the range 80-94 km in 10    solstice seasons, they then turned to be easterly. the reversal is at about 95 km (Smith, 2012). The zonal winds observed by meteor radar at a nearby site Maui (20.7°N, 156.4°W) (see figure 1a in Li et al., 2012) showed further that the westerly winds prevailed throughout the 80-100 km altitude range in the summer months from May to August. This provides us the opportunity to compare our GW $E_p$ climatology shown in Fig. 9a with the mean zonal wind climatology shown in the Fig. 1a of Li et al. (2012) or the upper panel of the Fig.3 in Smith (2012) season to season and altitude to altitude. Here we focus on 15    the altitude range 85-100 km.

Firstly, the mean zonal winds have a dominated SAO with westerly winds prevailing in solstice seasons and easterly winds (or weak westerly winds) prevailing in equinoxes seasons, meanwhile, our GW $E_p$ has a SAO with minima in winter and summer and maxima during equinoxes. Secondly, the easterly winds are much larger (or the westerly winds are much smaller) in the altitude range 85-95 km around vernal equinox than around autumn equinox, which corresponds to the fact 20    that the magnitude of GW $E_p$ in spring is significantly greater than that in autumn. This correlation is also verified by the fitted curve in Fig. 9b. The maximum of $E_p$ at vernal equinox with a value of 404 J. kg$^{-1}$ is a factor of 1.3 larger than the second maximum of 319 J. kg$^{-1}$ at autumn equinox. Thirdly, the largest westerly winds near 90 km in June matches perfectly with the minimal $E_p$ at almost the same altitude and at almost the same time. Fourthly, the decrease of easterly winds with altitude near 85 km during equinoxes is accordance to the strong but decreasing GW $E_p$ with altitude in almost the same 25    altitude range and seasons. Fifthly, the transition of mean zonal winds from decreasing westerly winds to increasing easterly winds above 96 km before middle April and After July corresponds well with the increasing $E_p$ in the same altitude range and seasons. These five features provide strong evidence to an indeed pronounced correlation between the local mean zonal wind field and the lidar observed GW $E_p$. This correlation agrees perfectly with the connection of wind and GW in the middle atmosphere demonstrated by Lindzen (1981). It means that the seasonal variations of the GW activity in the MLT 30    region at this site is determined by the selective filtering of GWs by the strong tropical zonal wind SAO at the same region.

It is noted that the GWEP increases instead of being constant with altitude as expected for freely propagating linear waves above 98 km in Spring and above 96 km in Autumn as shown in Fig. 10 and Fig. 9a. We notice that the corresponding regions of increasing GWEP enhancement with altitude in Fig. 9a associated well with the regions of increasing zonal wind

shears (in magnitude) in Figures 1a and 1b of Li et al. (2012), which suggests that conversion of energy contained in the mean wind into the energy of perturbation due to the increasing dynamical instability should be a most possible reason for the enhancement of GWEP near 100 km altitude in equinox seasons.

## 6 Summary and Conclusion

The first complete range-resolved climatology of potential energy in the tropical mesopause region is present using 11 years long nocturnal temperature measurements by the K Doppler lidar over the Arecibo Observatory. The mean temperature $\overline{T}$, the square of the Brunt-Väisälä frequency $N^2$ and the potential energy of perturbations associated with GWs are estimated with high accuracy and resolution from the temperature data. The main characteristics of the observations are as follows.

1. Mesospheric TILs occur in the altitude range 90-95 km in most months except from September to December.

2. The GW potential energy per unit volume (in $J.m^{-3}$) reduces in the altitude range 85-97 km in all seasons. Close relationship exists between the reduction of GW potential energy and the TILs. This provides strong support to the mechanism of the TIL formation in the mesopause region.

3. The seasonal variations of GW potential energy show clear SAO at most altitudes. The maxima occur in spring and autumn and the minima occur during solstices. AO and some high order oscillations still play important roles. The harmonic

fitted GW potential energy with the annual mean plus the AO and SAO is compared to the MLT wind field in the tropical region as published by Li et al. (2012) and Smith (2012), There is indeed a pronounced altitudinal and temporal correlation between them. This suggests that the seasonal variation of GW activity should be determined mainly by the local wind field through the influence of critical level filtering of GW by the background wind.

4. The GWEP increases near 100 km latitude in equinox seasons instead of being constant with altitude as expected for

freely propagating linear waves. This most possibly caused by the conversion of energy contained in the mean wind into the energy of perturbations due to dynamical instability.

*Competing interests*. The authors declare that they have no conflict of interest.

*Acknowledgments.* The study is supported by NSFC grants 41474128, 61771352 and NSF grant AGS-1744033. The Arecibo Observatory is operated by The University of Central Florida under a cooperative agreement with the National Science Foundation (AST-1744119) and in alliance with Yang Enterprises and Ana G. Méndez -Universidad Metropolitana. The authors thank Dr. John Anthony Smith, Dr. Frank Djuth, Dr. Dave Hysell, Dr. Min-Chang Lee and Eframir Franco Diaz for

their help with the observations. In addition, we are grateful to the two anonymous reviewers and Editor Robert Hibbins for improving the paper with their helpful comments.

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

Table 1. Arecibo K lidar temperature data used in this study (Days/Hours) by month.

| month | Total D/H[a] | 2004 D/H | 2005 D/H | 2006 D/H | 2007 D/H | 2008 D/H | 2009 D/H | 2010 D/H | 2015 D/H | 2016 D/H | 2017 D/H |
|---|---|---|---|---|---|---|---|---|---|---|---|
| Jan. | 32/259 | 4/27 | | 4/44 | 2/14 | 6/34 | | 4/35 | | 7/62 | 5/43 |
| Feb. | 12/102 | | 5/33 | | | | | | | 4/38 | 3/31 |
| Mar. | 20/136 | 2/9 | 10/74 | | 4/30 | 1/4 | 3/19 | | | | |
| Apr. | 22/152 | 1/5 | 14/103 | 3/16 | | 1/8 | 2/12 | | | | 1/8 |
| May | 6/40 | 2/12 | 2/18 | 2/10 | | | | | | | |
| Jun. | 9/63 | 3/21 | 2/16 | | | | 1/1 | | | 3/25 | |
| Jul. | 18/114 | 6/41 | 1/8 | 1/9 | | 5/26 | 5/30 | | | | |
| Aug. | 18/114 | 7/46 | 4/27 | 2/11 | 1/6 | 2/10 | | | | 2/14 | |
| Sep. | 14/109 | | 6/41 | 4/36 | | 1/8 | | | | 3/24 | |
| Oct. | 4/28 | 1/5 | 3/23 | | | | | | | | |
| Nov. | 18/155 | 1/2 | 5/46 | 4/31 | | | 1/9 | | 4/34 | 3/33 | |
| Dec. | 27/198 | 1/5 | 4/30 | 2/19 | | | 6/42 | 3/27[b] | 3/36 | 5/39 | |
| **Total** | **200/1470** | **28/173** | **56/419** | **22/176** | **7/50** | **16/90** | **18/113** | **7/62[c]** | **7/70** | **27/235** | **9/82** |

[a] D/H stands for Days/Hours; [b] Observed in 2003; [c] Including the observations in 2003.

**Table 2. Parameters of mean temperature, temperature variance, squared Brunt-Väisälä frequency and potential energy averaged between 87 and 97 km.**

| | Annual | Amplitude | | Phase (days) | | RMS |
|---|---|---|---|---|---|---|
| | Mean | 12-month | 6-month | 12-month | 6-month | Residual-σ |
| **Mean temperature (K)** | 188.7 | 3.6 | 1.8 | -34 | -57 | 5.5 |
| $\overline{N^2}$ $(10^{-4}s^{-2})$ | 4.37 | 0.09 | 0.12 | 160 | 22 | 0.37 |
| **Potential Energy (Jkg$^{-1}$)** | 351.8 | 55.9 | 42.1 | 119 | -89 | 141.6 |

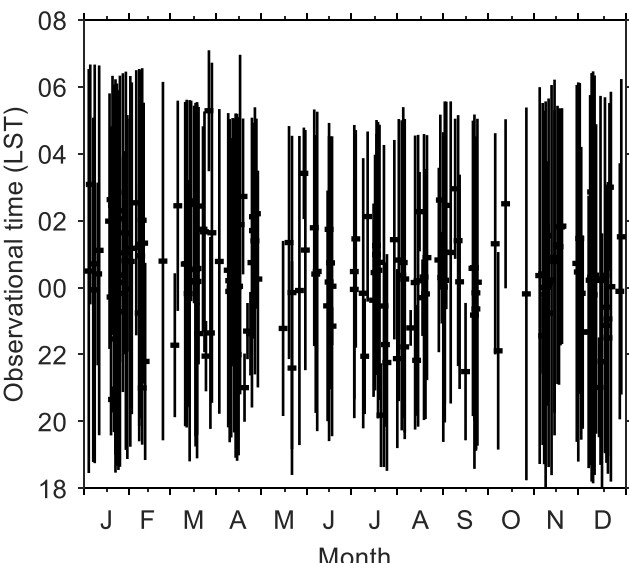

**Figure 1: Local time coverage of the used temperature data observed by the K-Doppler lidar at Arecibo from December 2003 to January 2010, and from November 2015 to April 2017.**

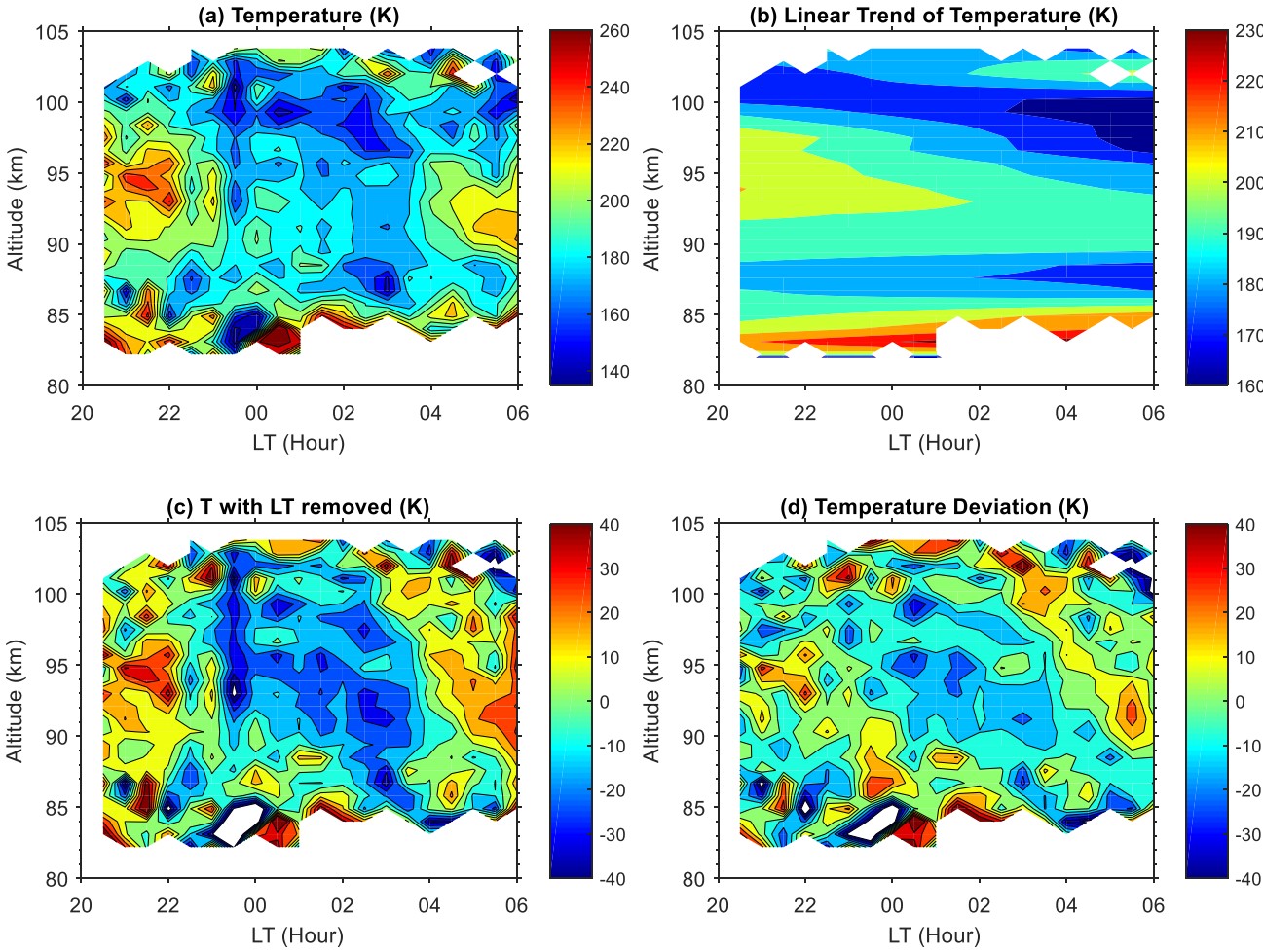

**Figure 2: Illustration of data processing for temperature on 18 December 2003: (a) K lidar measured temperature with 0.9 km and 30 minutes vertical and temporal resolutions, respectively; (b) linear trend of temperature in time; (c) temperature perturbations after removing the linear trend in time; and (d) temperature perturbations after removing the linear trend in time and the vertical mean.**

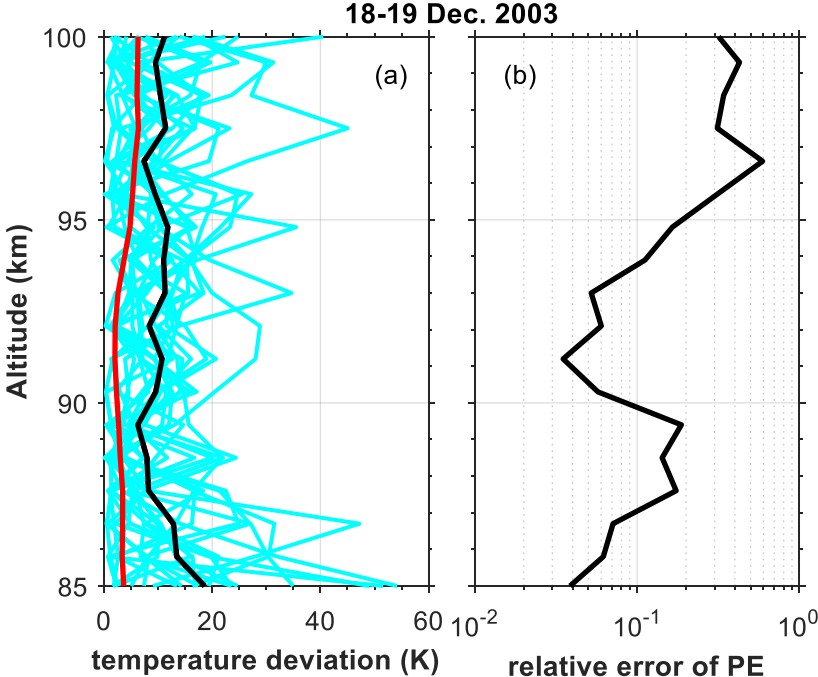

**Figure 3: (a)** Individual temperature deviation profiles on 18–19 December 2003. In addition, the mean statistical uncertainties of the measurements (red solid line) and the mean fluctuations at every altitude (black solid line) are shown; **(b)** relative error of potential energy at every altitude due to statistical uncertainty in temperature measurement.

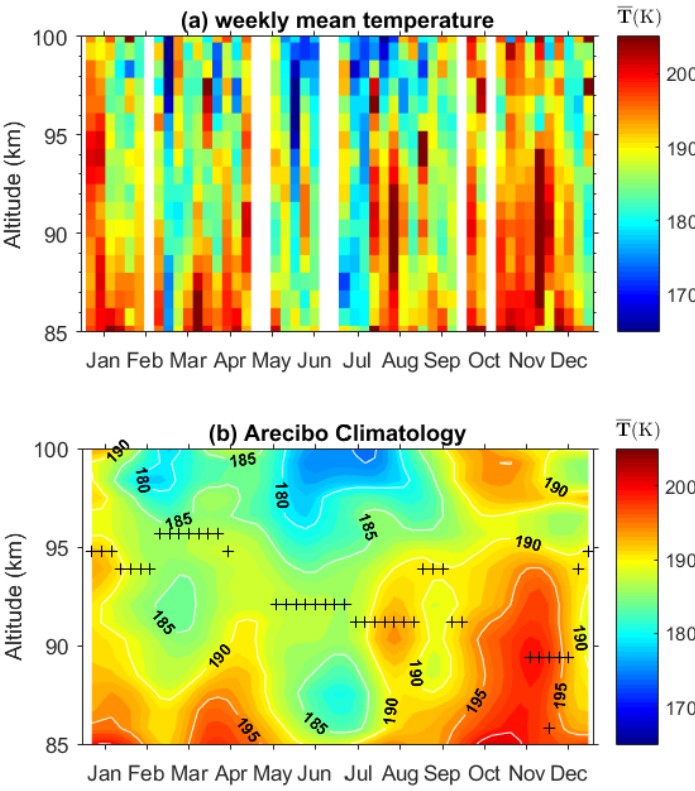

**Figure 4: (a) the weekly mean temperature profiles in the mesopause region at Arecibo. (b) temperature climatology obtained by first applying a harmonic fit to the data shown in (a) and adding the residuals smoothed by using a hamming window with lengths of 4 weeks in time and 2.7 km in vertical dimension.**

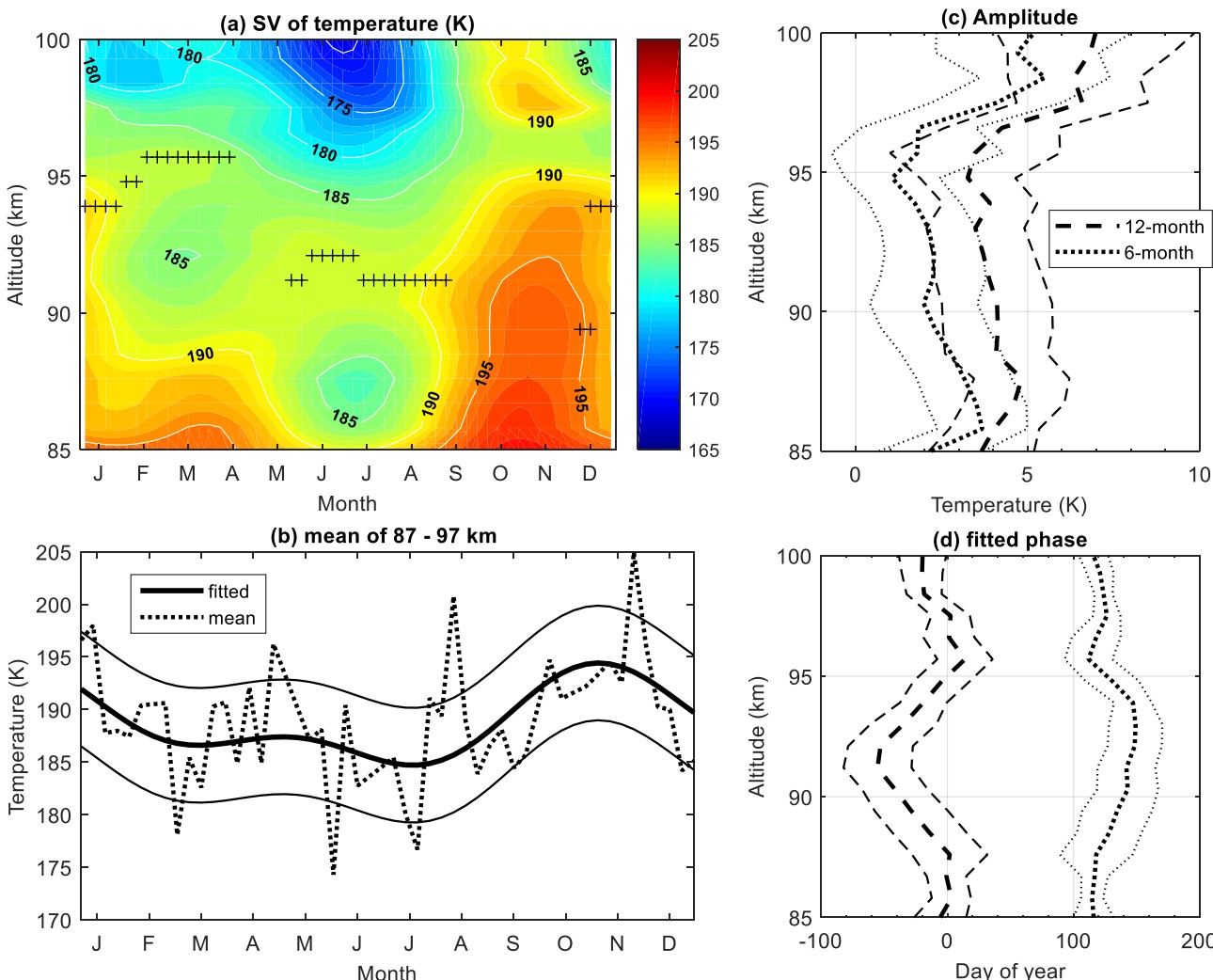

**Figure 5: (a)** Seasonal variations of the harmonic fitted nocturnal temperature plotted versus altitude and month, the crosses represent the altitude of temperature inversion layer, **(b)** observed (dotted curve) and harmonic fitted (thick solid curve) mean temperature between 87 and 97 km, the width between the thin solid curves and the thick solid curve is 1standard error (1σ), **(c)** 12-month (dashed curve) and 6-month (dotted curve) amplitudes and their 1 σ deviations (thin lines) **(d)** 12-month (solid curve) and 6-month (dotted curve) phases and their 1 σ deviations (thin lines).

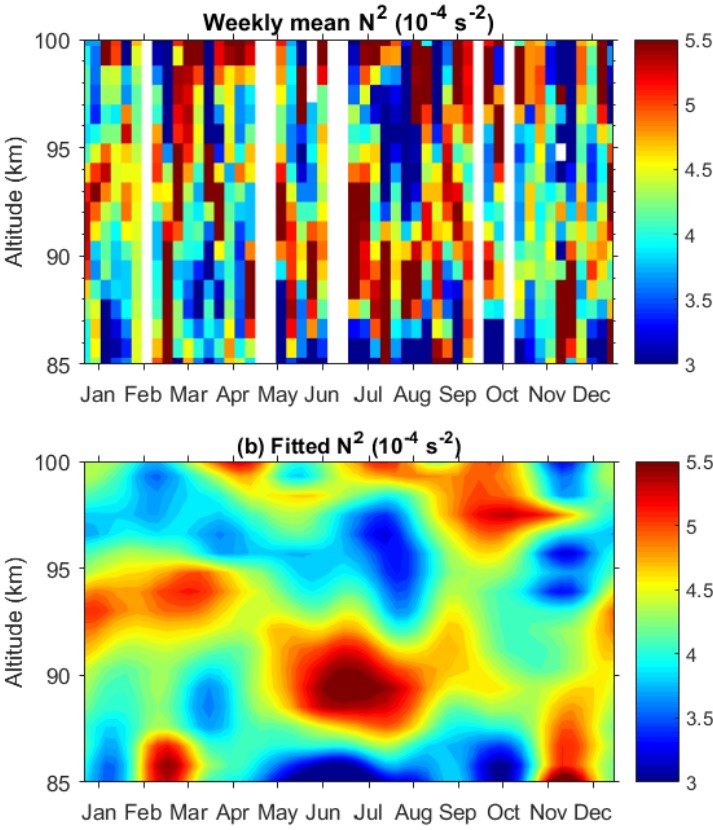

**Figure 6: same as Figure 4 but for $N^2$**

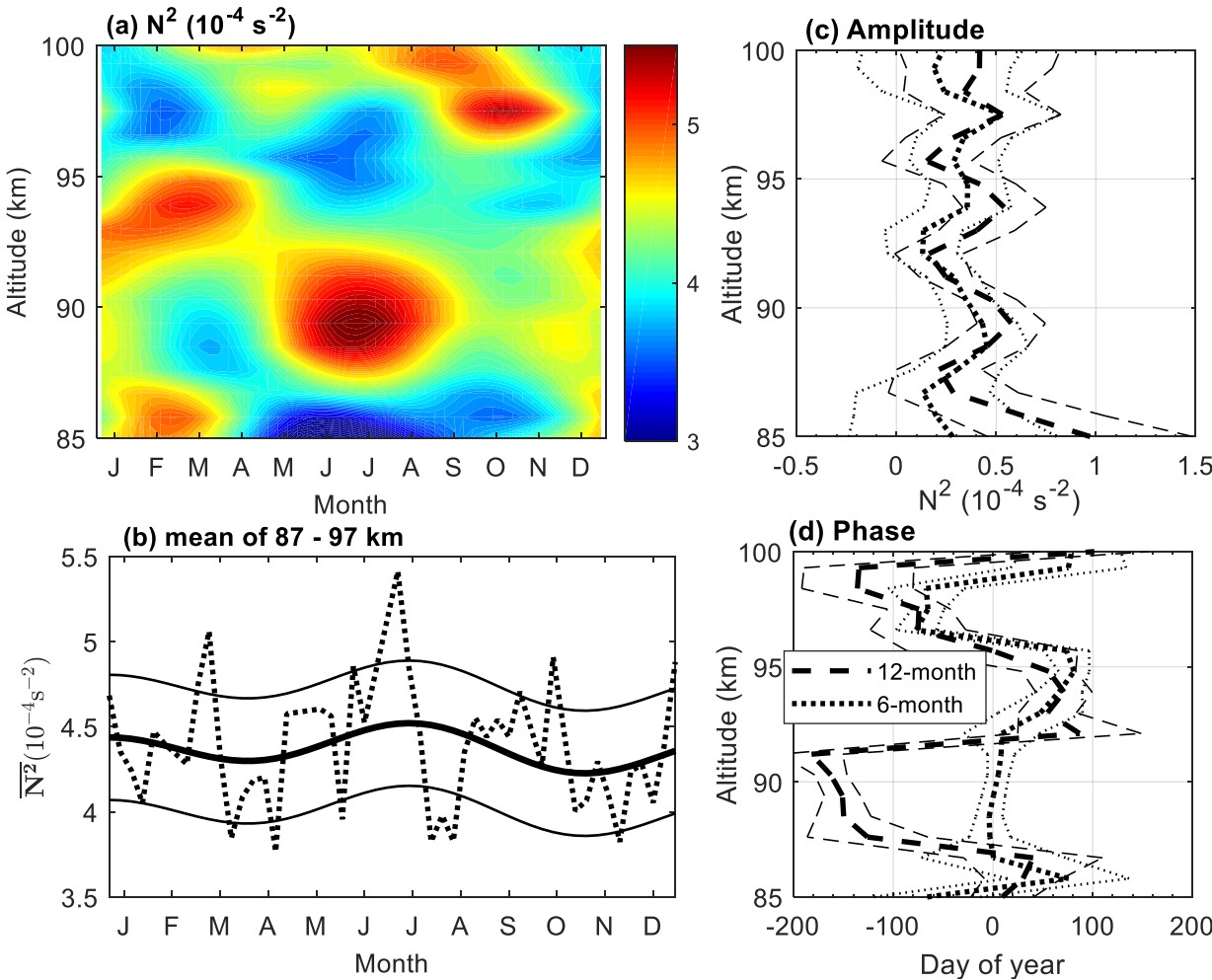

Figure 7: (a) Seasonal variations of the harmonic fitted squared Brunt-Väisälä frequency $N^2$ plotted versus altitude and month, (b) observed (dotted curve) and harmonic fitted (thick solid curve) mean $N^2$ between 87 and 97 km, the width between the thin solid curves and the thick solid curve is $1\sigma$, (c) 12-month (thick dashed curve) and 6-month (thick dotted curve) amplitudes and their $1\,\sigma$ deviations (thin lines), (d) 12-month (thick dashed curve) and 6-month (thick dotted curve) phases and their $1\,\sigma$ deviations (thin lines).

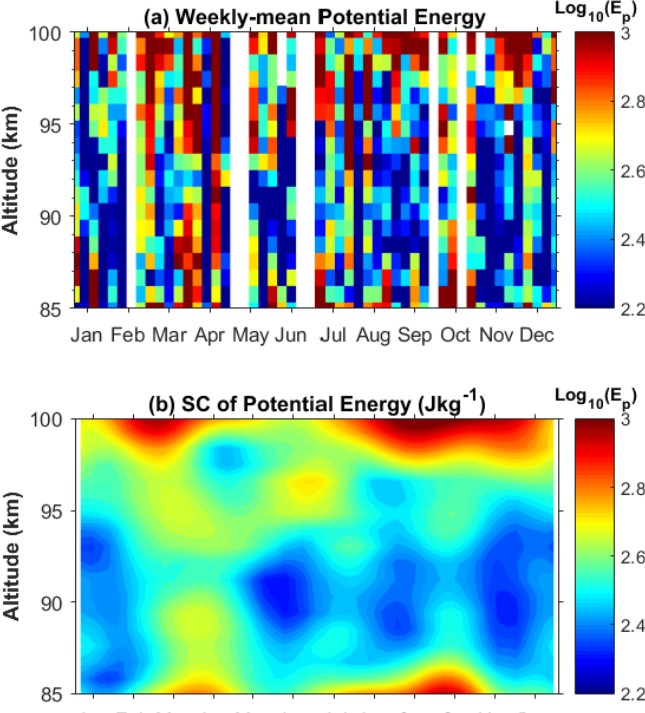

**Figure 8: same as Figure 4 but for the GW potential energy.**

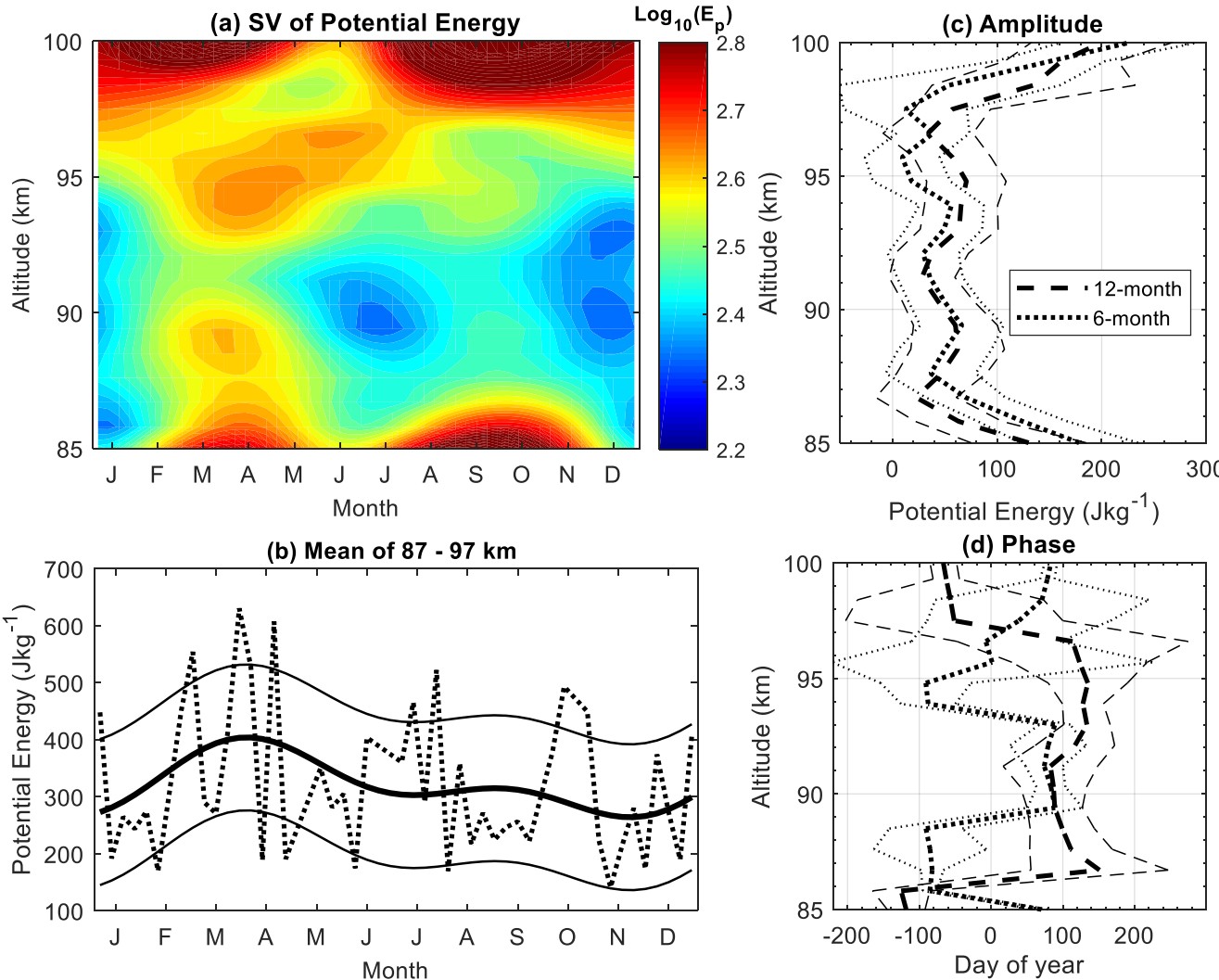

**Figure 9: (a) Seasonal variations of the harmonic fitted potential energies plotted versus altitude and month, (b) observed (dotted curve) and harmonic fitted (thick solid curve) mean potential energy between 87 and 97 km, the width between the thin solid curves and the thick solid curve is 1σ, (c) 12-month (thick dashed line) and 6-month (thick dotted line) amplitudes and their 1 σ deviations (thin lines), (d) 12-month (thick dashed line) and 6-month (thick dotted line) phases and their 1 σ deviations (thin lines).**

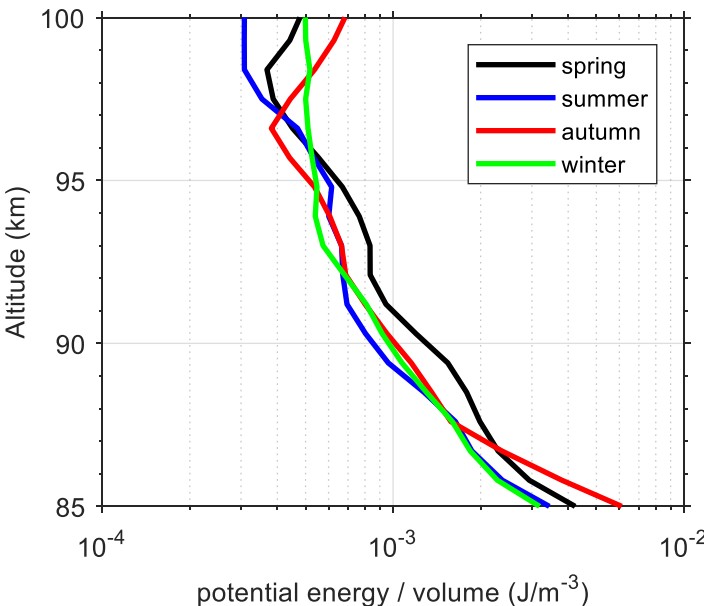

**Figure 10: Vertical profiles of the potential energy per unit volume (in J. m$^{-3}$) averaged over spring (13 weeks mean centred at vernal equinox, black line), summer (13 weeks mean centred at summer solstice, blue line), autumn (13 weeks mean centred at autumn equinox, red line), winter (13 weeks mean centred at winter solstice, green line).**