# Peer review of "Long-term Lidar Observations of the Gravity Wave Activity near the Mesopause at Arecibo"

_Atmospheric Chemistry and Physics, 2018_

## Referee Comment (RC1) · Anonymous Referee #1 · 30 Aug 2018

General comments:

This paper shows the extended climatology of temperature and potential energy density above Arecibo using lidar data. My main comment about the paper is that the work on gravity wave activity is not a major part of the paper despite it's title. I would like to see included at least one comparison with other gw lidar studies in the mesopause region (regardless of latitude) to see how their results compare in terms of seasonal variation or magnitude of gw activity observed. Perhaps also an expansion of the GW section by also looking at the year to year variation of GW PE if the authors feel it is appropriate and are not planning on doing this for a future paper.

Specific comments:

[Figure]

Page 3, line 1 – What do you mean by the conservation of GW potential energy? This needs a clearer explanation.

Page 3, lines 2-3 – you need to include more detail into why these studies show that more attention should be paid to the mesosphere in terms of gw parameterisations. What do your results in this paper offer that will help improve these parametrisations?

Page3, line 9 – "transforming" is not the right word here, I think you mean changes. Also, what is the change in the mean zonal wind above 80 km in the tropical region? This needs to be explained.

Page 4 – line 1 – Can you please include a reason as to why there is a 5 year gap in the dataset. Is the data from two different K lidars? Was the one lidar broken?

Page 4, equations 1 and 2 – why have you used the EP equations for temperature from Vincent et al and not used the one that Mze et al (2014) use in their lidar studies?

Page 4, line 21 – please include a brief description of the procedure for calculating T' rather than just pointing at a reference.

Page 4, line 25 – Doesn't applying this Hamming window alter again the minimum period and wavelength gravity waves that you will be able to detect? This will make the values you state at the start of section 2 invalid. Please address this in the text.

Page 4, lines 27-29 – what model is referred to here? I suspect it's the harmonic fit used in the Friedman and Chu paper you reference but it's not clear at all. More detail on what exactly is being done here and why is needed in the text.

Page 5, line 19 – why is the secondary peak insignificant? Surely it is just not as large, why does that make it statistically insignificant?

Page 5, line 21 – I have compared Figs 6 and 7 in the Friedman and Chu paper (F&C) with your Figure 2 and yes the annual variation is similar but there are also large differences that need to be explained. In your Fig 2a the vertical temperature structure is

different to that shown in F&C, with you showing warmer temperatures around March and October/November that are more extensive that those in the F&C paper. Also the semi-annual phase and amplitudes they show are quite different to yours in Fig 2c&d (which is expected as the SAO you show is different). The question needs to be asked as to why your climatology (which includes the data used in the F&C paper) is showing such differences. Are you using the exact same method as F&C? If not, when you perform your analysis on the same section of data as used in F&C do the results agree? Are there one or two years which have this warmer vertical structure and that is influencing the results in your paper? You need to explain why you are seeing a different structure to other results which use part of the same dataset.

Page 6, line 25 – you need to show an example of the seasonal cycle of the zonal winds to which you refer to in the paper

Figure 2a – can you plot the MIL you refer to on the text on the figure

Figure 4a – it might be easier to compare with other sites/lidar gw studies if you plot the lognormal of the GW PE.

Technical corrections:

Abstract – the phrase "potential energy of the temperature fluctuations" is not correct. You are using the temperature fluctuations to determine the potential energy density of the gravity wave field, i.e. the gravity wave activity levels. Please change so that it is correct.

Page 2, line 21 – change "are" to "have been"

Page 2, line 34 – eminent is not the right word to use here, do you mean evident?

Page 2, line 34 – again I don't think you mean to use "almost", "also" would make more sense.

Page 3, line 5 – "These researches" should be replaced with something like: "The

studies"

Page 3, line 18-19 – The sentence "the vertical structures of SAO and AO in these parameters and their relationships are exhibits" does not make sense, please rephrase.

Page 4, equations 1 and 2 – The overbar on the temperature indicates averaging over altitude, please include this in your description of the variables.

Page 4, line 22 – replace 0.5h with 30 minutes (or replace 30 minutes with 0.5h in the other instance in the paper). Try to be consistent with how you refer a time interval.

---

## Referee Comment (RC2) · Anonymous Referee #2 · 14 Sep 2018

Xianchang Yue et al. report about temperature soundings obtained with a potassium resonance lidar at Arecibo (18°N). Overall, 1451 h of data are obtained between December 2003 and April 2017, with good data coverage especially in the first three years. From this data set the seasonal variations of temperatures and their variability are derived, with emphasis mainly on the AO and SAO. There are only few temperature data sets available from the tropical mesopause region, and these data are a worthwhile contribution. The paper is well written and the Figures are of good quality. Unfortunately the whole reasoning is partly incomplete and digs not very deep into the data. Examples are given below. Overall, I recommend revision and extension of the manuscript.

General comments:

[Figure]

- The authors describe in the Discussion a relation between the wind field as published by Garcia et al. (1997) and Smith (2012) and the observed variation of GWPED. While there is indeed a pronounced altitudinal and temporal correlation, the paper lacks a description of the mechanism that relates the GW activity and zonal wind velocity. All statements are true, but remain vague and unspecific. The interpretation seems to imply pure zonal propagation of the waves, but the lidar data contain waves of all directions. Is the westerly wind between 60 and 70 km taken into account that may filter a lot of the eastward propagating GW?

- The authors do not show any kind of raw data, i.e. mean temperature profiles of a single night or even examples for temperature variability (T'). The lowest level figures are fitted AO and SAO, and it remains open, how representative they are. The authors state that these are the most important variations, but the large variability of the mean values in 2b/3b/4b and the "random phases" (l. 8/1 on page 8) seem to contradict. Therefore I recommend to provide also examples for T and T', as well as unfitted seasonal variations of all relevant quantities.

- The calculation of T' should be described in much more detail. This is the most crucial point for the interpretation, and a reference to Gardner and Liu 2007 is not sufficient. What are the main points in the retrieval? How does incomplete sampling influence the results? How are tides removed from the fluctuations? How is the increasing uncertainty at the layer edges acknowledged?

Specific comments:

l. 1/14-15: I think it is not a unique feature that $N^2$ maximizes below/at an inversion. This is just the result of the temperature increase with altitude (dT/dz is large and positive).

l. 1/26: I suggest to replace "usually" by "often" as there are also stratospheric sources, secondary waves etc.
l. 2/9: There is a logical break. I suggest writing "The stratospheric SAO leads to a seasonal variation of filtering of the upward propagating waves, which results in a specific seasonal variation of GW activity in the mesosphere."

l. 2/34-3/1: Sounds odd. Suggestion "Mze (. . .) observed a nearly undamped propagation of GW . . .".

l. 3/10: What is meant by "transforming of the mean zonal wind"?

l. 3/26 and l. 3/31: The reference to Yue at al. 2017 is not appropriate here, because the cited paper mainly deals with K density data.

l. 4/4: The worst data coverage is (by chance?) right in the month of largest wave activity. This should be discussed shortly.

l. 4/11-13: No. The ratio of kinetic and potential energy is a function of the intrinsic period of the wave (and the Coriolis parameter). From temperature soundings only the potential energy can be calculated, but not the total energy, because the intrinsic period is generally unknown.

l. 5/8: Please motivate the choice of this altitude interval. Especially $N^2$ is strongly varying in this range, and the phases of T and $N^2$ precess.

l. 5/13: Is the inversion also visible in the raw data or is it a result of the fit? If it is real, it should be discussed in more detail because it may strongly affect the propagation of GW. If it is not discussed here, some reference should be made.

l. 5/15: It remains open from this Figure, whether the mesopause could also be above 100 km in Sep-Dec.

l. 5/20 and Fig. 2: It would be helpful to have contour lines to assess the similarity between the data sets.

l. 5/28: This is essentially expected if $N^2$ is calculated from the mean temperature data set. See above.

l. 6/1-2: Please explain why it is worth to note this. My impression is that any kind of instability would be eliminated by building the temperature composite and applying an AO/SAO fit. Is the mesopause region more stable above Arecibo than somewhere else? If so, please explain and provide a reference.

l. 6/4-5: Consider plotting the phase of the SAO shifted by 180 d between 96 and 99 km (-80 will be +100). It may look nicer. Is the variability of the AO phase really unexpected (being the derivative of another property)? How does the variable phase affect the conclusions of the paper?

l. 6/8/9: Do I understand correctly that the fitted T and Nˆ2 are used as the mean value to calculate T'? This would be a significant difference to Gardner&Liu. Furthermore, using the fitted time series will affect the seasonal variation of the derived GW activity. Please clarify.

l. 7/9-10: I actually do not understand this sentence. Temperature enhancement from dissipating GW? Secondary waves? How relates a strength in temperature variation to a wind velocity?

l. 7/13: How much of the increase above 97 km is due to the increasing uncertainty at the edge of the K layer or by a (potential) deviation of the fitted data from the original data?

l. 8/1: I do not agree that the phases of T and Nˆ2 are "random", even if they are less constant with altitude. A random phase would imply that AO/SAO are not relevant oscillations in this altitude range.

l. 8/5-6: Please be more specific: Which processes do you expect that relate the GW activity to the seasonal wind variation in your data and what evidence you have?

l. 8/9-10: This statement is rather vague. Please be more precise.

Figure 2-4: Please provide error bars for fit amplitudes and phases.

Figure 4: I recommend plotting the GWPED per volume. For linear propagation this should be conserved. The strong increase of GWPED close to 100 km would be less pronounced.

Technical comments / Typos:

l. 3/19: "exhibits" should read "exhibited"

l. 4/8: "evenly" should read "even"

l. 5/2: "is" should read "are"

l. 6/14: It is Fig. 4a

---

## Author Comment (AC1) · 23 Sep 2018

Response to the Referee #1 Interactive comment on "Long-term Lidar Observations of the Gravity Wave Activity near the Mesopause at Arecibo" by Xianchang Yue et al. Anonymous Referee #1 Response: Thank you for your constructive and kind comments.

General comments: This paper shows the extended climatology of temperature and potential energy density above Arecibo using lidar data. My main comment about the paper is that the work on gravity wave activity is not a major part of the paper despite it's title. I would like to see included at least one comparison with other gw lidar studies in the mesopause region (regardless of latitude) to see how their results

compare in terms of seasonal variation or magnitude of gw activity observed. Perhaps also an expansion of the GW section by also looking at the year to year variation of GW PE if the authors feel it is appropriate and are not planning on doing this for a future paper. Response: there are some reports about the seasonal variations of temperature variances around mesopause in the literatures, we will add some comparisons about temperature variances in the revised manuscript after finishing the reprogramming of the data processing program.

Specific comments: Page 3, line 1 – What do you mean by the conservation of GW potential energy? This needs a clearer explanation. Reponse: Thank you for pointing out this improper sentence, the corresponding sentence has been rephrased by "Mzé et al. (2014) observed a nearly undamped propagation of GW in summer in the low mesosphere". Page 3, lines 2-3 – you need to include more detail into why these studies show that more attention should be paid to the mesosphere in terms of gw parameterizations. What do your results in this paper offer that will help improve these parameterizations? Response: Thank you for this comment, this sentence has been revised as "Since the effects of GW in the numerical climate and weather prediction models are usually represented simply by parameterization (Kim et al., 2003). there are still large discrepancies between model and measurement results (Geller et al., 2013). Therefore, more attention should be paid to the GW parameterization about these kind of observations in the upper mesosphere and mesopause region to improve the model results." Page3, line 9 – "transforming" is not the right word here, I think you mean changes. Also, what is the change in the mean zonal wind above 80 km in the tropical region? This needs to be explained. Response: "transition" is a proper word to replace "transforming". The changes in the mean zonal wind near mesopause in the tropical region has been introduced in the 3rd paragraph of the introduction section. Page 4 – line 1 – Can you please include a reason as to why there is a 5 year gap in the dataset. Is the data from two different K lidars? Was the one lidar broken? Response: In the time from 2011 to 2015 the lidar building was upgraded with an extension for the telescopes which were in a hut until this time. Page4, equations 1 and 2 –why have

you used the EP equations for temperature from Vincent et al and not used the one that Mze et al (2014) use in their lidar studies? Response: The difference between equation (1) in this manuscript and the equation (8) in Mze et al (2014) is that $(T'/T \bar{I}\check{E})^2$ in this (1) is replaced by atmosphere variance in that (8). The temperature inversion by an K Doppler lidar near the mesopause region uses the resonance scatter signal of K atoms, while the atmosphere variance estimation by a Rayleigh lidar from 30 to 80 km uses the Rayleigh scatter signal of atmosphere molecules which is taken as part of the background noise of K Doppler lidar.

Page 4, line 21 – please include a brief description of the procedure for calculating T' rather than just pointing at a reference. Response: The brief description added is as the following "For each night of observation, data points with photon noise errors larger than 10 K in temperature are discarded first. The linear trend in time is then subtracted from temperature profiles at each altitude to compute the temperature perturbations, perturbations exceeding three standard deviations from the nightly mean are discarded. Finally the vertical mean is subtracted from each temperature perturbation profile." Page 4, line 25 – Doesn't applying this Hamming window alter again the minimum period and wavelength gravity waves that you will be able to detect? This will make the values you state at the start of section 2 invalid. Please address this in the text. Response: Thank you for this question. We made a mistake in the writing. We have not applied this Hamming window on the temperature perturbation Tˆ'. "The weekly composite night data of $\hat{A}\acute{r}T$ and Tˆ' are" have been updated to "The weekly composite night data of $\hat{A}\acute{r}T$ is" after we checked the data processing MATLAB procedure. Page 4, lines 27-29 – what model is referred to here? I suspect it's the harmonic fit used in the Friedman and Chu paper you reference but it's not clear at all. More detail on what exactly is being done here and why is needed in the text. Response: Yes, it is just a harmonic fit, we have added the following texts and equation in the revised manuscript: "The equation of the model is as following: $\Psi(z,t)=\Psi_0(z)+A_{12}(z)\cos[2\pi/(365/7)(t-\varphi_{12}(z))]+A_6(z)\cos[4\pi/(365/7)(t-\varphi_6(z))]$ (3) where $\Psi(z,t)$ is the value of a weekly mean parameter at altitude z and week t, expressed in week of the

year (1-52), $\Psi_0$ (z) is the annual mean, $A_n$ (z) and $\varphi_n$ (z) (n=6,12) are the amplitude and phase of the -month oscillation, respectively." Page 5, line 19 – why is the secondary peak insignificant? Surely it is just not as large, why does that make it statistically insignificant? Response: We have used an improper word here. 'An insignificant secondary' has been updated to "A secondary" in the revised version. Page 5, line 21 – I have compared Figs 6 and 7 in the Friedman and Chu paper (F&C) with your Figure 2 and yes the annual variation is similar but there are also large differences that need to be explained. In your Fig 2a the vertical temperature structure is different to that shown in F&C, with you showing warmer temperatures around March and October/November that are more extensive that those in the F&C paper. Also the semi-annual phase and amplitudes they show are quite different to yours in Fig 2c&d (which is expected as the SAO you show is different). The question needs to be asked as to why your climatology (which includes the data used in the F&C paper) is showing such differences. Are you using the exact same method as F&C? If not, when you perform your analysis on the same section of data as used in F&C do the results agree? Are there one or two years which have this warmer vertical structure and that is influencing the results in your paper? You need to explain why you are seeing a different structure to other results which use part of the same dataset. Response: Thank you for pointing out the difference between these two work. The differences are caused by three reasons. The first and the key point is the quality control to calculate the temperature perturbation in this study. Figure 1 in the appendixes of this response shows the fitted result without quality control. It is quite consistent with the Figure 6 in F&C. The second reason is the much more extensive data set from year 2003 to 2017 covering a whole solar period here. The last reason is the harmonic fit model is in term of week here, while it was in term of month in Friedan and Chu (2007).

Page 6, line 25 – you need to show an example of the seasonal cycle of the zonal winds to which you refer to in the paper Response: Thank you for this suggestion. We have added "(see e.g., Fig. 3 in Garcia et al. 1997; Fig. 3 in Smith 2012), the monthly mean HRDI equatorial zonal wind showed that, the easterly winds were prevailing in

equinoxes seasons near 80 km altitude. They then decreased with altitude from 80 km above and turned to increase above ∼ 92 km, while the westerly winds prevailed in in the range 80-94 km in solstice seasons, they then turn to be easterly. the reversal is at about 95 km (Smith, 2012). Therefore, the zonal winds are low or zero around 92 km altitude in tropical region. The zero-wind lines will enhance damping or dissipating of zonal propagating gravity wave with low to moderate phase speed.' between "In retrospect to the scientific literature in SAO studies" and "The features of gravity wave potential energy . . . . . .". Figure 2a – can you plot the MIL you refer to on the text on the figure Response: Figure 4a – it might be easier to compare with other sites/lidar gw studies if you plot the lognormal of the GW PE. Response:

Technical corrections: Abstract – the phrase "potential energy of the temperature fluctuations" is not correct. You are using the temperature fluctuations to determine the potential energy density of the gravity wave field, i.e. the gravity wave activity levels. Please change so that it is correct. Response: Thank you for pointing out this error. Phrase 'potential energy of the temperature fluctuation' has been changed to 'potential energy derived from the temperature data' in the revision Page 2, line 21 – change "are" to "have been" Response: It is changed. Page 2, line 34 – eminent is not the right word to use here, do you mean evident? Response: "evident" is proper here, and 'eminent' is replaced with 'evident'. Page 2, line 34 – again I don't think you mean to use "almost", "also" would make more sense. Response: Thank you for this comment, we have rewritten this sentence as "Mzé et al. (2014) observed a nearly undamped propagation of GW in summer in the low mesosphere." Page 3, line 5 – "These researches" should be replaced with something like: "The studies" Response: It is replaced. Page 3, line 18-19 – The sentence "the vertical structures of SAO and AO in these parameters and their relationships are exhibits" does not make sense, please rephrase. Response: since this sentence is not necessary, we omitted it in the revised version. Page 4, equations 1 and 2 – The overbar on the temperature indicates averaging over altitude, please include this in your description of the variables. Response: It is included according to this suggestion. Page 4, line 22

– replace 0.5h with 30 minutes (or replace 30 minutes with 0.5h in the other instance in the paper). Try to be consistent with how you refer a time interval. Response: "0.5h" has been replaced with 30 minutes here.

Please also note the supplement to this comment:
https://www.atmos-chem-phys-discuss.net/acp-2018-731/acp-2018-731-AC1-supplement.pdf
* * *
[Figure]

**Fig. 1.**

**Supplement:**

[revised manuscript text omitted]

---

## Author Comment (AC2) · 23 Sep 2018

Response to the comments from Referee #2.
Xianchang Yue et al. report about temperature soundings obtained with a potassium resonance lidar at Arecibo (18°N). Overall, 1451 h of data are obtained between December 2003 and April 2017, with good data coverage especially in the first three years. From this data set the seasonal variations of temperatures and their variability are derived, with emphasis mainly on the AO and SAO. There are only few temper-

ature data sets available from the tropical mesopause region, and these data are a worthwhile contribution. The paper is well written and the Figures are of good quality. Unfortunately the whole reasoning is partly incomplete and digs not very deep into the data. Examples are given below. Overall, I recommend revision and extension of the manuscript.

Response: Thank you for your hard work for review and for your evaluation of the manuscript. We will try to make it more valuable through revision.

General comments: The authors describe in the Discussion a relation between the wind field as published by Garcia et al. (1997) and Smith (2012) and the observed variation of GWPED. While there is indeed a pronounced altitudinal and temporal correlation, the paper lacks a description of the mechanism that relates the GW activity and zonal wind velocity. All statements are true, but remain vague and unspecific. The interpretation seems to imply pure zonal propagation of the waves, but the lidar data contain waves of all directions. Is the westerly wind between 60 and 70 km taken into account that may filter a lot of the eastward propagating GW?

Response:

- The authors do not show any kind of raw data, i.e. mean temperature profiles of a single night or even examples for temperature variability (T'). The lowest level figures are fitted AO and SAO, and it remains open, how representative they are. The authors state that these are the most important variations, but the large variability of the mean values in 2b/3b/4b and the "random phases" (l. 8/1 on page 8) seem to contradict. Therefore I recommend to provide also examples for T and T', as well as unfitted seasonal variations of all relevant quantities. Response: The unfitted composited weekly mean temperature profiles are shown in Fig. 2 in the revised manuscript. As the programs to deal with the data in this study are lost in a recent update of my computer, I've not reprogrammed them all out till this deadline of commend. Therefore, I have to provide other quantities in the later revised manuscript.

-The calculation of T' should be described in much more detail. This is the most crucial point for the interpretation, and a reference to Gardner and Liu 2007 is not sufficient. What are the main points in the retrieval? How does incomplete sampling influence the results? How are tides removed from the fluctuations? How is the increasing uncertainty at the layer edges acknowledged? Response: The procedure to calculate temperature perturbation is described in detail in the revised manuscript according to this commend.

Specific comments: l. 1/14-15: I think it is not a unique feature that NËĘ2 maximizes below/at an inversion. This is just the result of the temperature increase with altitude (dT/dz is large and positive). Response: thank you for your comment, I learn it. This point has been deleted.

l. 1/26: I suggest to replace "usually" by "often" as there are also stratospheric sources, secondary waves etc. Response: "usually" has been replaced by 'often'.

l. 2/9 there is a logical break. I suggest writing "The stratospheric SAO leads to a seasonal variation of filtering of the upward propagating waves, which results in a specific seasonal variation of GW activity in the mesosphere." Response: thank you for this correction, and we have replace this sentence with your suggestion.

l. 2/34-3/1: Sounds odd. Suggestion "Mze (...) observed a nearly undamped propagation of GW ...". Response: Thank you for this suggestion, and this sentence has been updated accordingly.

l. 3/10: What is meant by "transforming of the mean zonal wind"? Response: "transforming" should be "transition", means that the direction of the mean zonal wind changes.

l. 3/26 and l. 3/31: The reference to Yue at al. 2017 is not appropriate here, because the cited paper mainly deals with K density data. Response: This reference has been deleted in at these two places.

l. 4/4: The worst data coverage is (by chance?) right in the month of largest wave activity. This should be discussed shortly. Response: We give a discussion in the section of "Seasonal Variation of the Mean Temperature" by comparing the lidar data to the SABER observed mean zonal temperature reported by Xu et al. (2007).

l. 4/11-13: No. The ratio of kinetic and potential energy is a function of the intrinsic period of the wave (and the Coriolis parameter). From temperature soundings only the potential energy can be calculated, but not the total energy, because the intrinsic period is generally unknown. Response: Thank you for your opinion, these sentence has been omitted in the revised version.

l. 5/8: Please motivate the choice of this altitude interval. Especially $N\ddot{E}_2$ is strongly varying in this range, and the phases of T and $N\ddot{E}_2$ precess. Response: The temperature error due to photon noise is usually less than 5 K in the raw data located in this altitude interval 87-97 km because of the rather larger K density. But it is in a risk of diminishing the feature of seasonal variation for a strongly varying parameter in term of altitude such as $N\hat{\ }2$. We demonstrate the motivation and discuss the risk around this sentence in the revised manuscript.

l. 5/13: Is the inversion also visible in the raw data or is it a result of the fit? If it is real, it should be discussed in more detail because it may strongly affect the propagation of GW. If it is not discussed here, some reference should be made. Response: Yes, the inversion is visible in the raw data. We have discussed the temperature inversion layer and its effects on the propagation of GW in section 4.4 in detail.

l. 5/15: It remains open from this Figure, whether the mesopause could also be above 100 km in Sep-Dec. Response: Yes, it does.

l. 5/20 and Fig. 2: It would be helpful to have contour lines to assess the similarity between the data sets. Response: I'm trying to reprogram all the data process program.

l. 5/28: This is essentially expected if $N\ddot{E}_2$ is calculated from the mean temperature

data set. See above. Response: Yes, it is not a finding but only an effect.

l. 6/1-2: Please explain why it is worth to note this. My impression is that any kind of instability would be eliminated by building the temperature composite and applying an AO/SAO fit. Is the mesopause region more stable above Arecibo than somewhere else? If so, please explain and provide a reference. Response: It is not worthy to note, consequently, this sentence is omitted.

l. 6/4-5: Consider plotting the phase of the SAO shifted by 180 d between 96 and 99 km (-80 will be +100). It may look nicer. Is the variability of the AO phase really unexpected (being the derivative of another property)? How does the variable phase affect the conclusions of the paper? Response: I'm trying to reprogram all the data process program. It is going to be dealt with in another week. I have no further idea about how the variable phase affects the conclusions.

l. 6/8/9: Do I understand correctly that the fitted T and NË₂ are used as the mean value to calculate T'? This would be a significant difference to Gardner&Liu. Furthermore, using the fitted time series will affect the seasonal variation of the derived GW activity. Please clarify. Response: No, the nightly mean temperature is used to calculate T'. I just wanted to write a sentence connecting the preceding and the following, unfortunately, it is not correctly written. This sentence is omitted.

l. 7/9-10: I actually do not understand this sentence. Temperature enhancement from dissipating GW? Secondary waves? How relates a strength in temperature variation to a wind velocity? Response: I mean that the smaller of the zonal mean zonal wind amplitude, the smaller the GW amplitude. But I can't discuss it further, it is omitted.

l. 7/13: How much of the increase above 97 km is due to the increasing uncertainty at the edge of the K layer or by a (potential) deviation of the fit data from the original data? Response: the quantitative analysis is not given now.

l. 8/1: I do not agree that the phases of T and NË₂ are "random", even if they are

less constant with altitude. A random phase would imply that AO/SAO are not relevant oscillations in this altitude range. Response: the conclusion we drawn is subjective, the "are random" is updated to "vary" in this sentence.

l. 8/5-6: Please be more specific: Which processes do you expect that relate the GW activity to the seasonal wind variation in your data and what evidence you have? Response: These two sentences has been updated to "The seasonal variations and vertical structures of E_p relate closely to bot that of the HRDI zonal winds and that of the DW1 tide amplitude at 20°N reported in the literature. These relationships agree well with the effect of spectral filter due to lower atmosphere winds and the critical level encountering of GW accounting for the formation mechanism of upper mesosphere TIL."

l. 8/9-10: This statement is rather vague. Please be more precise. Response: this statement is omitted.

Figure 2-4: Please provide error bars for fit amplitudes and phases. Response: I'm trying to reprogram all the data process program.

Figure 4: I recommend plotting the GWPED per volume. For linear propagation this should be conserved. The strong increase of GWPED close to 100 km would be less pronounced. Response: I'm trying to reprogram all the data process program.

Technical comments / Typos: l. 3/19: "exhibits" should read "exhibited" response: thank you for pointing out this wrong spelling.

l. 4/8: "evenly" should read "even" Response: It is corrected. l. 5/2: "is" should read "are" Response: It is corrected.

l. 6/14: It is Fig. 4a Response: It is corrected.

[Figure]

Please also note the supplement to this comment:
https://www.atmos-chem-phys-discuss.net/acp-2018-731/acp-2018-731-AC2-supplement.pdf

---

## Editor Comment (EC1) · R. E. Hibbins (Editor) · 28 Sep 2018

Before preparing and submitting a revised manuscript would the authors please provide a response to referee #1's specific comments relating to figures 2a and 4a, and referee #2's first general comment. For clarity, these comments are reproduced below:

Referee #1

Figure 2a – can you plot the MIL you refer to on the text on the figure

Figure 4a – it might be easier to compare with other sites/lidar gw studies if you plot the lognormal of the GW PE.

Referee #2

[Figure]

The authors describe in the Discussion a relation between the wind field as published by Garcia et al. (1997) and Smith (2012) and the observed variation of GWPED. While there is indeed a pronounced altitudinal and temporal correlation, the paper lacks a description of the mechanism that relates the GW activity and zonal wind velocity. All statements are true, but remain vague and unspecific. The interpretation seems to imply pure zonal propagation of the waves, but the lidar data contain waves of all directions. Is the westerly wind between 60 and 70 km taken into account that may filter a lot of the eastward propagating GW?

---

## Author Comment (AC3) · 10 Oct 2018

Response to Editor Hibbins Interactive comment on "Long-term Lidar Observations of the Gravity Wave Activity near the Mesopause at Arecibo" by Xianchang Yue et al. R. E. Hibbins (Editor) robert.hibbins@ntnu.no Before preparing and submitting a revised manuscript would the authors please provide a response to referee#1's specific comments relating to figures 2a and4a, and referee #2's first general comment. For clarity, these comments are reproduced below: Referee #1 Figure 2a – can you plot the MIL you refer to on the text on the figure Response: The MILs have been plotted on Figure 2a. Fig.2 has been updated in the revised version and is attached to this reply. Figure 4a – it might be easier to compare with other sites/lidar gw studies if you plot the lognormal of the GW PE. Response:

The GW PE in the Figure 4a and 4c of the original version are now plotted by the log-normal in Figure 5a and 5c in the revised version. And the corresponding Fig. 5 is updated in the revised version and is attached to this reply. Referee #2 The authors describe in the Discussion a relation between the wind field as published by Garcia et al. (1997) and Smith (2012) and the observed variation of GWPED. While there is indeed a pronounced altitudinal and temporal correlation, the paper lacks a description of the mechanism that relates the GW activity and zonal wind velocity. All statements are true, but remain vague and unspecific. The interpretation seems to imply pure zonal propagation of the waves, but the lidar data contain waves of all directions. Is the westerly wind between 60 and 70 km taken into account that may filter a lot of the eastward propagating GW? Response: Thank you for reproduced this constructive and kind comments. We have checked the relationship between the climatology of gravity wave potential energy and that of the background wind in the low-latitude mesopause region, and give some clear and specific statement. The updated description in the revised version is as following: "Here we also want to check the relation between our observed GW activity and the wind direction and/or wind speed. Some scientific literatures reported studies about seasonal variation of mean zonal wind in the mesopause region in low-latitude region (see e.g., Fig. 3 in Garcia et al. 1997; Fig. 3 in Smith 2012). This provides us the opportunity to compare our GW $E\_p$ climatology shown in Fig. 5a with the mean zonal winds climatology shown in the upper panel of the Fig.3 in Smith (2012) season to season and altitude to altitude. Here we focus on the altitude range 85-100 km. Firstly, the mean zonal winds have a dominated semiannual oscillation with westerly winds prevailing in solstice seasons and eastly winds prevailing in equinoxes seasons, meanwhile, our GW $E\_p$ has a semiannual oscillation with minima in winter and summer and with maxima during equinoxes. Secondly, the easterly winds are much larger in the altitude range 85-95 km around vernal equinox than around autumn equinox, which corresponds to the fact that the magnitude of GW $E\_p$ in spring is significantly greater than that in autumn. This is also verified by the fitted curve in Fig. 5b. The maximum of $E\_p$ at vernal equinox with a value of 404 J.ãĂŰkgãĂŮˆ(-1)

is a factor of 1.3 larger than the second maximum of 319 J.ãĂŰkgãĂŮˆ(-1) at autumn equinox. Thirdly, the largest westerly winds near 90 km in June matches perfectly with the minimum E_p at almost the same altitude range and in almost the same period. Fourthly, the zero wind line near and above 95 km altitude throughout a whole year is accordance to the almost equal E_p near 97 km in all seasons. Fifthly, the transition of mean zonal winds to easterly above 96 km throughout the whole year corresponds well with the overall increase of E_p in the same altitude range. These five features provide strong evidence to a definite relationship between the mean zonal wind direction and wind speed and the GW E_p. This relationship agrees perfectly with the connection of wind and GW in the middle atmosphere demonstrated by Lindzen (1981)."

Response to some other comments of reviewer #1 and #2 In this reply, we also want to reply some comments of reviewer #1 and #2 that was not responded well in the last reply. Reviewer #1 General comments: This paper shows the extended climatology of temperature and potential energy density above Arecibo using lidar data. My main comment about the paper is that the work on gravity wave activity is not a major part of the paper despite it's title. I would like to see included at least one comparison with other gw lidar studies in the mesopause region (regardless of latitude) to see how their results compare in terms of seasonal variation or magnitude of gw activity observed. Perhaps also an expansion of the GW section by also looking at the year to year variation of GW PE if the authors feel it is appropriate and are not planning on doing this for a future paper. Response: We have done comparisons to the gravity wave potential energy observations in the mesopause region at other stations. These comparisons are added in the discussion section of the revised version. The contexts are as following: "We point out a semi-annual cycle of GW E_p with maximum in spring and minimum in summer and a second maximum in autumn and a second minimum in winter in the altitude range 87-97 km. The maximum of the GW E_p alters to autumn below 87 km and above 97 km altitude. These results agree with the observations at other low-latitude station. Gavrilov et al. (2003) studied the GW seasonal variations by using Medium-Frequency (MF) radar observation over Hawaii (22°N, 160°W). They found a

semiannual variation of GW with the maximum intensity at the equinoxes above 83 km, the mean zonal wind had also a mainly semiannual variation in this altitude range. The seasonal variations of GW activities at low-latitude stations are different to those obtained from lidar observations at other latitude stations in the upper mesosphere (Mzé et al., 2014; Rauthe et al., 2006, 2008). Rauthes et al. (2008) provided the seasonal variations of GW $E\_p$ at a 54°N latitude station by using a 6-years of lidar temperature observations from 1 to 105 km. They showed an annual-dominated variation of GW $E\_p$ with the maximum in winter and the minimum in summer in the mesopause region. Mzé et al. (2014) reported a semi-annual variation of GW $E\_p$ with maxima in winter and in summer and minima during the equinoxes in the upper mesosphere (∼75.5 km) by using Rayleigh lidar observations from 1996 to 2012 at a mid-latitude station (∼44°N). They showed that the maximum of $E\_p$ was about 144 JãĂŰ.kgãĂŮˆ(-1) on average at 75.5 km in August while the minimum of $E\_p$ is about a factor of 2.5 smaller than the maximum. The factor of ratio between the maximum and the minimum is obviously larger than that of 1.5 in the altitude range 87-97 km at Arecibo."

Reviewer #2 Figure 4: I recommend plotting the GWPED per volume. For linear propagation this should be conserved. The strong increase of GWPED close to 100 km would be less pronounced. Response: We have added a figure (Fig. 6 in the revised version) to show the GW potential energy per unit volume (in J.mˆ(-3)). This figure is attached to this reply and the damp of potential energy is clear seen in this figure. The emoticon to the figure in the revised is "Figure 6: Vertical profiles of the potential energy per unit volume (in J.mˆ(-3)) averaged over spring (13 weeks centred at vernal equinox, black line), summer (13 weeks centred at summer solstice, blue line), autumn (13 weeks centred at autumn equinox, red line), winter (13 weeks centred at winter solstice, green line).". The description about this figure in the revised version is as following: "To learn in depth the dissipation of GW in the mesopause region at Arecibo, we multiplied the harmonic fitted $E\_p$ with the air density taken from the CIRA-86 reference atmosphere [Fleming et al., 1990], and average every 13 weekly profiles centering at each equinox or solstice. The resulted 4 profiles of the

potential energy per unit volume (in J.mˆ(-3)) are plotted in Fig. 6. If GWs propagate upward without energy dissipation, the lines of energy per unit volume would be vertical. Therefore, the four left-sloping lines in Fig. 6 indicate that the damps of GW potential energy occur below ∼95 km in all seasons. The damp of GW potential energy in the mesosphere had been reported by lidar observations at other latitude stations (e. g. Mzé et al., 2014; Rauthe et al., 2008). Both observations of Mzé et al. (2014) and Rauthe et al. (2008) indicate dissipation of GW E_p throughout the mesosphere in all seasons. It is noticed that the green line almost keeps vertical above 94 km which indicate that the GW potential energies are almost conserved in this altitude range in winter. The slops of the other three lines turn to be positive above ∼97 km. The transition of potential energy per volume from decrease to increase with altitude also occurred in the study of lidar observed GW activity at the 54°N latitude station (Rauthe et al. 2008) but with the transition altitude being above ∼93 km.

Please also note the supplement to this comment:
https://www.atmos-chem-phys-discuss.net/acp-2018-731/acp-2018-731-AC3-supplement.pdf
* * *
[Figure]

**Fig. 1.**

**Supplement:**

[revised manuscript text omitted]

- Ratnam, M.V., Tetzlaff, G., and Jacobi, C.: Global and Seasonal Variations of Stratospheric Gravity Wave Activity Deduced from the CHAMP/GPS Satellite, J. Atmos. Sci., 61, 1610-1620, 2004.
   Rauthe, M., Gerding, M., and Lübken, F.-J.: Seasonal changes in gravity wave activity measured by lidars at mid-latitudes, Atmos. Chem. Phys., 8, 6775-6787, https://doi.org/10.5194/acp-8-6775-2008, 2008.
- Sivakumar, V., Rao, P. B., and Bencherif, H.: Lidar observations of middle atmospheric gravity wave activity over a low-latitude site (Gadanki, 13.5°N, 79.2°E), Ann. Geophys., 24, 823–834, https://doi.org/10.5194/angeo-24-823-2006, 2006.
  Smith, A.: Global Dynamics of the MLT, Surv. Geophys., 33, 1177-1230, 2012.
  Tsuda, T., Nishida, M., Rocken, C., and Ware, R. H. A global morphology of gravity wave activity in the stratosphere

revealed by the GPS occultation data (GPS/MET), J. Geophys. Res.-Atmos., 105, 7257-7273, https://doi.org/10.1029/1999JD901005, 2000.

30 Vincent, R. A., Allen, S. J., and Eckermann, S. D.: Gravity Wave Processes: Their Parameterization in Climate Models, NATO ASI Series, Gravity wave parameters in the lower stratosphere, K. Hamilton, 150, Springer,7–25, 1997. Xu, J., Smith, A. K., Yuan, W., Liu, H.-L., Wu, Q., Mlynczak, M. G., and Russell III, J. M.; Global structure and long-term variations of zonal mean temperature observed by TIMED/SABER, J. Geophys. Res., 112, D24106, https://doi.org/10.1029/2007JD008546, 2007.

|--------------------------------------------------------------------------------|

Roman, 10 磅, 英语(英国)

Xu, J., A. K. Smith, H.-L. Liu, W. Yuan, Q. Wu, G. Jiang, M. G. Mlynczak, and J. M. Russell III, Estimation of the equivalent Rayleigh friction in mesosphere/lower thermosphere region from the migrating diurnal tides observed by TIMED, J. Geophys. Res., 114, D23103, doi:10.1029/2009JD012209, 2009.

Yuan, T., Heale, C.J., Snively, J.B., Cai, X., Pautet, P.-D., Fish, C., ... Mitchell, N. J.: Evidence of dispersion and refraction
of a spectrally broad gravity wave packet in the mesopause region observed by the Na lidar and Mesospheric Temperature Mapper above Logan, Utah, J. Geophys. Res.-Atmos., 121, 579–594, https://doi.org/10.1002/2015JD023685, 2016.

Yue, X., Zhou, Q., Raizada, S., Tepley, C., and Friedman, J.: Relationship between mesospheric Na and Fe layers from simultaneous and common-volume lidar observations at Arecibo, J. Geophys. Res.-Atmos., 118, 905–916, https://doi.org/10.1002/jgrd.50148, 2013.

10 Yue, X., Zhou, Q., Yi, F., Friedman, J., Raizada, S., and Tepley, C.: Simultaneous and common-volume lidar observations of K/Na layers and temperature at Arecibo Observatory (18°N, 67°W), J. Geophys. Res.-Atmos., 121, 8038-8054, https://doi.org/10.1002/2015JD024494, 2016.

Yue, X., Friedman, J. S., Wu, X., and Zhou, Q. H.: Structure and seasonal variations of the nocturnal mesospheric K layer at Arecibo, J. Geophys. Res.-Atmos., 122, 7260-7275. https://doi.org/10.1002/2017JD026541, 2017.

15 Yue, X. and Yi, F. Sci. Propagation of gravity wave packet near critical level, China Ser. E-Technol. Sci., 48, 538. https://doi.org/10.1360/102005-24, 2005.

20

|---|-------------------------------------------------------------------------------|

|------------------|---------------------------------------------------------------------------------------------------|

|       | Total            | 2004   | 2005   | 2006   | 2007 | 2008  | 2009   | 2010              | 2015 | 2016   | 2017 |
|-------|------------------|--------|--------|--------|------|-------|--------|-------------------|------|--------|------|
| month | D/H a | D/H    | D/H    | D/H    | D/H  | D/H   | D/H    | D/H               | D/H  | D/H    | D/H  |
| Jan.  | 21/253           | 4/27   |        | 4/44   | 2/14 | 6/34  |        | 4/35              |      | 7/62   | 5/43 |
| Feb.  | 7/101            |        | 5/33   |        |      |       |        |                   |      | 4/38   | 3/31 |
| Mar.  | 20/135           | 2/9    | 10/74  |        | 4/30 | 1/4   | 3/19   |                   |      |        |      |
| Apr.  | 21/149           | 1/5    | 14/103 | 3/16   |      | 1/8   | 2/12   |                   |      |        | 1/8  |
| May   | 6/45             | 2/12   | 2/18   | 2/10   |      |       |        |                   |      |        |      |
| Jun.  | 8/62             | 3/21   | 2/16   |        |      |       | 1/1    |                   |      | 3/25   |      |
| Jul.  | 18/112           | 6/41   | 1/8    | 1/9    |      | 5/26  | 5/30   |                   |      |        |      |
| Aug.  | 16/112           | 7/46   | 4/27   | 2/11   | 1/6  | 2/10  |        |                   |      | 2/14   |      |
| Sep.  | 12/107           |        | 6/41   | 4/36   |      | 1/8   |        |                   |      | 3/24   |      |
| Oct.  | 4/27             | 1/5    | 3/23   |        |      |       |        |                   |      |        |      |
| Nov.  | 11/152           | 1/2    | 5/46   | 4/31   |      |       | 1/9    |                   | 4/34 | 3/33   |      |
| Dec.  | 16/196           | 1/5    | 4/30   | 2/19   |      |       | 6/42   | 3/27 b | 3/36 | 5/39   |      |
| Total | 198/1451         | 28/171 | 56/417 | 22/163 | 7/50 | 16/88 | 18/113 | 7/62°             | 7/70 | 27/235 | 9/82 |

Table 1. Arecibo K lidar temperature data used in this study (Days/Hours) by month.

 $^{\rm a}$  D/H stands for Days/Hours;  $^{\rm b}$  Observed in 2003;  $^{\rm c}$  Including the observations in 2003.

5

Table 2. Parameters of mean temperature, temperature variance, squared Brunt-Väisälä frequency and potential energy averaged between 87 and 97 km.

|                                                      | Annual | Amplitude |         | Phase    | RMS     |          |
|------------------------------------------------------|--------|-----------|---------|----------|---------|----------|
|                                                      | Mean   | 12-month  | 6-month | 12-month | 6-month | Residual |
| Mean temperature (K)                                 | 188.7  | 3.6       | 1.8     | -34      | -57     | 5.5      |
| $\overline{N^2}$ (10 -4 s -2 ) | 4.37   | 0.09      | 0.12    | 160      | 22      | 0.37     |
| Potential Energy (Jkg -1 )                | 351.8  | 55.9      | 42.1    | 119      | -89     | 141.6    |

Figure 1: Local time coverage of the used temperature data observed by the K-Doppler lidar at Arecibo from December 2003 to January 2010, and from November 2015 to April 2017.

Figure 2: (a) Seasonal variations of night time temperature plotted versus altitude and month, (b) seasonal variation of mean temperature between 87 and 97 km, (c) 12-month (dashed) and 6-month (dotted) amplitudes and (d) 12-month (solid) and 6-month (dotted) phases.

---

## Author Comment (AC4) · 10 Oct 2018

The Fig.2 and Fig. 4 (now Fig.5 in the revised version) is updated, they are attached to this reply.
* * *
[Figure]

**(a) fitted temperature (K)**

**(b) mean of 87 - 97 km**

**(c) fitted amplitude**

- - - 12-month
· · · · · 6-month

**(d) fitted phase**

**Fig. 1.** the updated Fig.2

[Figure]

(a) fitted Potential Energy

(b) Mean of 87 - 97 km

(c) Amplitude

(d) Phase

**Fig. 2.** the updated Fig.5

---

## Author Response (AR1)

5 Anonymous Referee #1 Received and published: 30 August 2018

Author's response: Thank you for your constructive and kind comments.

**Comments from referees:**

General comments: This paper shows the extended climatology of temperature and potential energy density above Arecibo using lidar data. My main comment about the paper is that the work on gravity wave activity is not a major part of the paper

10 despite it's title. I would like to see included at least one comparison with other gw lidar studies in the mesopause region (regardless of latitude) to see how their results compare in terms of seasonal variation or magnitude of gw activity observed. Perhaps also an expansion of the GW section by also looking at the year to year variation of GW PE if the authors feel it is appropriate and are not planning on doing this for a future paper.

Author's response: Thank reviewer #1 for this constructive comment. We have done comparisons to the gravity wave potential energy observations in the mesopause region at other stations. These comparisons are added in the discussion section of the revised version. The contexts are as following:

**Author's changes in manuscript**: *"We point out a semi-annual cycle of GW Ep with maximum in spring and minimum in summer and a second maximum in autumn and a second minimum in winter in the altitude range 87-97 km. The maximum of the GW Ep alters to autumn below 87 km and above 97 km altitude. These results agree with the observations at other low-*

- 20 latitude station. Gavrilov et al. (2003) studied the GW seasonal variations by using Medium-Frequency (MF) radar observation over Hawaii (22°N, 160°W). They found a semiannual variation of GW with the maximum intensity at the equinoxes above 83 km, the mean zonal wind had also a mainly semiannual variation in this altitude range. The seasonal variations of GW activities at low-latitude stations are different to those obtained from lidar observations at other latitude stations in the upper mesosphere (Mzé et al., 2014; Rauthe et al., 2006, 2008). Rauthes et al. (2008) provided the seasonal
- 25 variations of  $GW E_p$  at a 54°N latitude station by using a 6-years of lidar temperature observations from 1 to 105 km. They showed an annual-dominated variation of  $GW E_p$  with the maximum in winter and the minimum in summer in the mesopause region. Mzé et al. (2014) reported a semi-annual variation of  $GW E_p$  with maxima in winter and in summer and minima during the equinoxes in the upper mesosphere (~75.5 km) by using Rayleigh lidar observations from 1996 to 2012 at a midlatitude station (~44 ° N). They showed that the maximum of  $E_p$  was about 144 J.kg-1 on average at 75.5 km in August while the minimum of  $E_p$  is about a factor of 2.5 smaller than the maximum. The factor of ratio between the maximum and the minimum is obviously larger than that of 1.5 in the altitude range 87-97 km at Arecibo."

**Comments from referees:**

5 Specific comments:

Page 3, line 1 – What do you mean by the conservation of GW potential energy? This needs a clearer explanation.

Author's response: Thank you for pointing out this improper sentence, the corresponding sentence has been rephrased by "Mzé et al. (2014) observed a nearly undamped propagation of GW in summer in the low mesosphere".

**Comments from referees:**

10 Page 3, lines 2-3 – you need to include more detail into why these studies show that more attention should be paid to the mesosphere in terms of gw parameterizations. What do your results in this paper offer that will help improve these parameterizations?

Author's response: Thank you for this comment, this sentence has been revised as:

Author's changes in manuscript: "Since the effects of GW in the numerical climate and weather prediction models are usually represented simply by parameterization (Kim et al., 2003). there are still large discrepancies between model and measurement results (Geller et al., 2013). Therefore, more attention should be paid to the GW parameterization about these kind of observations in the upper mesosphere and mesopause region to improve the model results."

**Comments from referees:**

Page3, line 9 – "transforming" is not the right word here, I think you mean changes. Also, what is the change in the mean 20 zonal wind above 80 km in the tropical region? This needs to be explained.

Author's response: "transition" is a proper word to replace "transforming". The changes in the mean zonal wind near mesopause in the tropical region has been introduced in the 3rd paragraph of the introduction section.

**Comments from referees:**

Page 4 – line 1 – Can you please include a reason as to why there is a 5 year gap in the dataset. Is the data from two different K lidars? Was the one lidar broken?

Author's response: In the time from 2011 to 2015 the lidar building was upgraded with an extension for the telescopes which were in a hut until this time.

**Comments from referees:**

Page4, equations 1 and 2 – why have you used the EP equations for temperature from Vincent et al and not used the one that 5 Mze et al (2014) use in their lidar studies?

Author's response: The difference between equation (1) in this manuscript and the equation (8) in Mze et al (2014) is that  $\left(\frac{T'}{\bar{T}}\right)^2$  in this (1) is replaced by atmosphere variance in that (8). The temperature inversion by an K Doppler lidar near the mesopause region uses the resonance scatter signal of K atoms, while the atmosphere variance estimation by a Rayleigh lidar from 30 to 80 km uses the Rayleigh scatter signal of atmosphere molecules which is taken as part of the background noise of

10 K Doppler lidar.

**Comments from referees:**

Page 4, line 21 - please include a brief description of the procedure for calculating T' rather than just pointing at a reference.

Author's response: The brief description added is as the following:

Author's changes in manuscript: "For each night of observation, data points with photon noise errors larger than 10 K in temperature are discarded first. The linear trend in time is then subtracted from temperature profiles at each altitude to compute the temperature perturbations, perturbations exceeding three standard deviations from the nightly mean are discarded. Finally the vertical mean is subtracted from each temperature perturbation profile."

**Comments from referees:**

Page 4, line 25 – Doesn't applying this Hamming window alter again the minimum period and wavelength gravity waves
that you will be able to detect? This will make the values you state at the start of section 2 invalid. Please address this in the text.

Author's response: Thank you for this question. We made a mistake in the writing. We have not applied this Hamming window on the temperature perturbation T'. "*The weekly composite night data of*  $\overline{T}$  *and* T' *are*" have been updated to "*The weekly composite night data of*  $\overline{T}$  *is*" after we checked the data processing MATLAB procedure.

Page 4, lines 27-29 – what model is referred to here? I suspect it's the harmonic fit used in the Friedman and Chu paper you reference but it's not clear at all. More detail on what exactly is being done here and why is needed in the text.

Author's response: Yes, it is just a harmonic fit, we have added the following texts and equation in the revised manuscript:

**5 Author's changes in manuscript:**

"The equation of the model is as following:

$$\Psi(z,t) = \Psi_0(z) + A_{12}(z) \cos\left[\frac{2\pi}{365/7} \left(t - \varphi_{12}(z)\right)\right] + A_6(z) \cos\left[\frac{4\pi}{365/7} \left(t - \varphi_6(z)\right)\right]$$
(3)

where  $\Psi(z, t)$  is the value of a weekly mean parameter at altitude z and week t, expressed in week of the year (1-52),  $\Psi_0(z)$  is the annual mean,  $A_n(z)$  and  $\varphi_n(z)$  (n = 6,12) are the amplitude and phase of the -month oscillation, respectively."

**10 Comments from referees:**

Page 5, line 19 – why is the secondary peak insignificant? Surely it is just not as large, why does that make it statistically insignificant?

Author's response: We have used an improper word here. 'An insignificant secondary' has been updated to "A secondary" in the revised version.

**15 Comments from referees:**

Page 5, line 21 – I have compared Figs 6 and 7 in the Friedman and Chu paper (F&C) with your Figure 2 and yes the annual variation is similar but there are also large differences that need to be explained. In your Fig 2a the vertical temperature structure is different to that shown in F&C, with you showing warmer temperatures around March and October/November that are more extensive that those in the F&C paper. Also the semi-annual phase and amplitudes they show are quite

- 20 different to yours in Fig 2c&d (which is expected as the SAO you show is different). The question needs to be asked as to why your climatology (which includes the data used in the F&C paper) is showing such differences. Are you using the exact same method as F&C? If not, when you perform your analysis on the same section of data as used in F&C do the results agree? Are there one or two years which have this warmer vertical structure and that is influencing the results in your paper? You need to explain why you are seeing a different structure to other results which use part of the same dataset.
- 25 Author's response: Thank you for pointing out the difference between these two works. The differences are caused by three reasons.

Author's changes in manuscript: "The first and the key point is the lack of adding the smoothed residual temperature back to T(z,t) estimated by using (3) in this study. The second reason is the much more extensive data set from year 2003 to 2017 covering a whole solar period here. The last reason is the harmonic fit model is in term of week here, while it was in term of month in Friedan and Chu (2007)."

**5 Comments from referees:**

Page 6, line 25 - you need to show an example of the seasonal cycle of the zonal winds to which you refer to in the paper

Author's response: Thank you for this suggestion. We have added:

Author's changes in manuscript: "(see e.g., Fig. 3 in Garcia et al. 1997; Fig. 3 in Smith 2012), the monthly mean HRDI equatorial zonal wind showed that, the easterly winds were prevailing in equinoxes seasons near 80 km altitude. They then

10 decreased with altitude from 80 km above and turned to increase above ~ 92 km, while the westerly winds prevailed in in the range 80-94 km in solstice seasons, they then turn to be easterly. the reversal is at about 95 km (Smith, 2012). Therefore, the zonal winds are low or zero around 92 km altitude in tropical region. The zero-wind lines will enhance damping or dissipating of zonal propagating gravity wave with low to moderate phase speed' in the discussion.

**Comments from referees:**

15 Figure 2a – can you plot the MIL you refer to on the text on the figure

Author's response: The altitude of MILs has been plotted by black crosses in the revised Fig.3a.

**Comments from referees:**

Figure 4a – it might be easier to compare with other sites/lidar gw studies if you plot the lognormal of the GW PE.

Author's response: The GW PE is plotted in log-normal in the revised Fig. 5a.

**20 Comments from referees:**

Technical corrections:

Abstract – the phrase "potential energy of the temperature fluctuations" is not correct. You are using the temperature fluctuations to determine the potential energy density of the gravity wave field, i.e. the gravity wave activity levels. Please change so that it is correct.

Author's response: Thank you for pointing out this error. Phrase 'potential energy of the temperature fluctuation' has been changed to 'potential energy derived from the temperature data' in the revision

**Comments from referees:**

Page 2, line 21 - change "are" to "have been"

5 Author's response: It is changed in the revised version.

**Comments from referees:**

Page 2, line 34 – eminent is not the right word to use here, do you mean evident?

Author's response: "evident" is proper here, and 'eminent' is replaced with 'evident'.

**Comments from referees:**

10 Page 2, line 34 – again I don't think you mean to use "almost", "also" would make more sense.

Author's response: Thank you for this comment, we have rewritten this sentence as "Mzé et al. (2014) observed a nearly undamped propagation of GW in summer in the low mesosphere."

**Comments from referees:**

Page 3, line 5 - "These researches" should be replaced with something like: "The studies"

**15 Author's response: It is replaced.**

**Comments from referees:**

Page 3, line 18-19 – The sentence "the vertical structures of SAO and AO in these parameters and their relationships are exhibits" does not make sense, please rephrase.

Author's response: since this sentence is not necessary, we omitted it in the revised version.

**20 Comments from referees:**

Page 4, equations 1 and 2 – The overbar on the temperature indicates averaging over altitude, please include this in your description of the variables.

Author's response: It is included according to this suggestion.

Page 4, line 22 – replace 0.5h with 30 minutes (or replace 30 minutes with 0.5h in the other instance in the paper). Try to be consistent with how you refer a time interval.

7

Author's response: "0.5h" has been replaced with 30 minutes here.

**Response to the comments from Referee #2.**
Xianchang Yue et al. report about temperature soundings obtained with a potassium resonance lidar at Arecibo (18°N). Overall, 1451 h of data are obtained between December 2003 and April 2017, with good data coverage especially in the first three years. From this data set the seasonal variations of temperatures and their variability are derived, with emphasis mainly

10 on the AO and SAO. There are only few temperature data sets available from the tropical mesopause region, and these data are a worthwhile contribution. The paper is well written and the Figures are of good quality. Unfortunately the whole reasoning is partly incomplete and digs not very deep into the data. Examples are given below. Overall, I recommend revision and extension of the manuscript.

**Author's response**: Thank you for your hard work for review and for your evaluation of the manuscript. We will try to make it more valuable through revision.

**Comments from referees:**

**General comments:**

The authors describe in the Discussion a relation between the wind field as published by Garcia et al. (1997) and Smith (2012) and the observed variation of GWPED. While there is indeed a pronounced altitudinal and temporal correlation, the

20 paper lacks a description of the mechanism that relates the GW activity and zonal wind velocity. All statements are true, but remain vague and unspecific. The interpretation seems to imply pure zonal propagation of the waves, but the lidar data contain waves of all directions. Is the westerly wind between 60 and 70 km taken into account that may filter a lot of the eastward propagating GW?

**Author's response:**

- 25 Thank this reviewer for this valuable comment. I've made significant revision and extension in discussing the relation between the wind field and the observed variation of GWPED. These discussions are presented in section 5.3 of the revised version and will be listed in the following paragraph. There is indeed a pronounced altitudinal and temporal correlation between them. This promotes us to a statement that "*This suggests that the seasonal variation of GW activity should be determined mainly by the local wind field through the influence of critical level filtering of GW by the background wind*" in
- 30 the conclusion section. The revised discussions in section 5.3 are as following:

Author's changes in manuscript: "Here we also want to check the relation between our observed GW activity and the wind direction and/or wind speed. Some scientific literatures reported studies about seasonal variation of mean zonal wind in the tropical mesopause region (see e.g., Fig. 3 in Garcia et al. 1997; Fig. 3 in Smith 2012). The monthly mean HRDI equatorial zonal wind showed that, the easterly winds were prevailing in equinoxes seasons near 80 km altitude. They then decreased

- 5 with altitude from 80 km above and turned to increase above ~ 92 km, while the westerly winds prevailed in in the range 80-94 km in solstice seasons, they then turn to be easterly. the reversal is at about 95 km (Smith, 2012). This provides us the opportunity to compare our GW  $E_p$  climatology shown in Fig. 5a with the mean zonal winds climatology shown in the upper panel of the Fig.3 in Smith (2012) season to season and altitude to altitude. Here we focus on the altitude range 85-100 km. Firstly, the mean zonal winds have a dominated semiannual oscillation with westerly winds prevailing in solstice seasons
- 10 and easterly winds prevailing in equinoxes seasons, meanwhile, our  $GW E_p$  has a semiannual oscillation with minima in winter and summer and with maxima during equinoxes. Secondly, the easterly winds are much larger in the altitude range 85-95 km around vernal equinox than around autumn equinox, which corresponds to the fact that the magnitude of  $GW E_p$ in spring is significantly greater than that in autumn. This correlation is also verified by the fitted curve in Fig. 5b. The maximum of  $E_p$  at vernal equinox with a value of 404 J.kg-1 is a factor of 1.3 larger than the second maximum of 319
- 15  $J.kg^{-1}$  at autumn equinox. Thirdly, the largest westerly winds near 90 km in June matches perfectly with the minimum  $E_p$  at almost the same altitude and at almost the same time. Fourthly, the zero-wind line near 96 km altitude throughout a whole year is accordance to the almost equal  $E_p$  at almost the same altitude in all seasons. Fifthly, the transition of mean zonal winds to easterly winds above 96 km throughout the whole year corresponds well with the overall increase of  $E_p$  in the same altitude range. These five features provide strong evidence to an indeed pronounced correlation between the local mean
- 20 zonal wind field and the lidar observed GW Ep. This correlation agrees perfectly with the connection of wind and GW in the middle atmosphere demonstrated by Lindzen (1981). Correlation between GW potential energy and local winds has been suggested by Wright et al. (2016) in their multi-instrument GW measurements over Tierra del Fuego (54°S, 68°W), which was devoted to the Doppler shifting of waves into the observational filters of the instruments by these winds."

**Comments from referees:**

- 25 The authors do not show any kind of raw data, i.e. mean temperature profiles of a single night or even examples for temperature variability (T'). The lowest level figures are fitted AO and SAO, and it remains open, how representative they are. The authors state that these are the most important variations, but the large variability of the mean values in 2b/3b/4b and the "random phases" (l. 8/1 on page 8) seem to contradict. Therefore I recommend to provide also examples for T and T', as well as unfitted seasonal variations of all relevant quantities.
- 30 Author's response: The unfitted composited weekly mean temperature profiles are shown in Fig. 2 in the revised manuscript. As the inclusion of the other quantities would change the structure of the paper significantly, we would like to provide the pictures of these quantities and the picture of processing temperature perturbation as supplements.

The calculation of T' should be described in much more detail. This is the most crucial point for the interpretation, and a reference to Gardner and Liu 2007 is not sufficient. What are the main points in the retrieval? How does incomplete

5 sampling influence the results? How are tides removed from the fluctuations? How is the increasing uncertainty at the layer edges acknowledged?

Author's response: The procedure to calculate temperature perturbation is described in detail in the revised manuscript according to this commend. It is as the following:

Author's changes in manuscript: "It includes 4 steps. Step 1: for each night of observation, data points with photon noise

- 10 errors larger than 30 K in temperature are discarded. The value of 30 K is set based on the fact that the root mean square errors (RMSE) due to photon noise often reach to about 40-50 K near the edges (~80 km on the bottom and ~105 km on the top) of the temperature profile. Step 2: the linear trend in time at each altitude in the temperature profile is then subtracted to eliminate the potential biases associated with GWs with periods longer than about twice the observation period. Step 3: Perturbations exceeding three standard deviations from the nightly mean are discarded from the resulted temperature
- 15 perturbation series at each altitude to remove occasional outliers. Step 4: the vertical mean is subtracted from each temperature perturbation profile to eliminate the influences of the waves with vertical wave length longer than about twice the profile height range (~ 25 km). The resulted temperature perturbation profiles usually cover the height range 80-105 km. Therefore, the weekly composite nights of temperature, N2 and the consequential Ep usually cover the height range 80-105 km. To avoid the uncertainty of the analysis, we focus on the height range 85-100 km where the RMSE of each
- 20 instantaneous observed temperature usually less than 10 K."

**Comments from referees:**

Specific comments:

l. 1/14-15: I think it is not a unique feature that  $N^2$  maximizes below/at an inversion. This is just the result of the temperature increase with altitude (dT/dz is large and positive).

25 Author's response: thank you for your comment, I learn it. This point has been deleted.

**Comments from referees:**

1. 1/26: I suggest to replace "usually" by "often" as there are also stratospheric sources, secondary waves etc.
 Author's response: "usually" has been replaced by 'often'.

I. 2/9 there is a logical break. I suggest writing "The stratospheric SAO leads to a seasonal variation of filtering of the upward propagating waves, which results in a specific seasonal variation of GW activity in the mesosphere." **Author's response**: thank you for this correction, and we have replaced this sentence with your suggestion.

**5 Comments from referees:**

1. 2/34-3/1: Sounds odd. Suggestion "Mze (...) observed a nearly undamped propagation of GW ...". **Author's response**: Thank you for this suggestion, and this sentence has been updated accordingly.

**Comments from referees:**

1. 3/10: What is meant by "transforming of the mean zonal wind"?

10 Author's response: "transforming" should be "transition", means that the direction of the mean zonal wind changes.

**Comments from referees:**

1. 3/26 and 1. 3/31: The reference to Yue at al. 2017 is not appropriate here, because the cited paper mainly deals with K density data.

Author's response: This reference has been deleted at these two places.

**15 Comments from referees:**

4/4: The worst data coverage is (by chance?) right in the month of largest wave activity. This should be discussed shortly.
 Author's response: We give a discussion in the section of "Seasonal Variation of the Mean Temperature" by comparing the lidar data to the SABER observed mean zonal temperature reported by Xu et al. (2007). The discussion is as following:
 Author's changes in manuscript: "Notice that the warmest temperature occurs around October with shortest observation

- 20 times which reduce the confidence level of the harmonic fit. However, the observation times in both September and November are longer than 100 hours in more than 10 nights, they help to keep the confidence level of the harmonic fit. Moreover, the temperature structure shown in Fig. 3a agrees well with the temporal variations of the equatorial zonal mean temperature in the range 85-100 km observed by SABER (Xu et al., 2007). The amplitudes and phases of both SAO and AO observed by SABER at 20° N latitude had been shown by Xu et al. (2007) in their middle panels of Figure 10. Comparisons
- 25 show that the lidar observed phases of both SAO and AO shown in Fig. 3d agree with those obtained by SABER in the same altitude range. The SAO amplitude shown in Fig. 3c agree quite well with that observed by SABER in both magnitude and vertical structure. The lidar AO amplitude show similar vertical structure with that of SABER, but the magnitude of lidar AO amplitude is at least 1 K larger than that observed by SABER. The agreement between lidar and SABER observations gives us more confidence to use the lidar observed temperature data studying the GW activities in latter sections".

1. 4/11-13: No. The ratio of kinetic and potential energy is a function of the intrinsic period of the wave (and the Coriolis parameter). From temperature soundings only the potential energy can be calculated, but not the total energy, because the intrinsic period is generally unknown.

5 Author's response: Thank you for your opinion, these sentences have been omitted in the revised version.

**Comments from referees:**

1. 5/8: Please motivate the choice of this altitude interval. Especially  $N^2$  is strongly varying in this range, and the phases of T and  $N^2$  precess.

Author's response: The temperature error due to photon noise is usually less than 5 K in the raw data located in this altitude interval 87-97 km because of the rather larger K density. But it is in a risk of diminishing the feature of seasonal variation for a strongly varying parameter in term of altitude such as N^2. We demonstrate the motivation and discuss the risk around this sentence in the revised manuscript. In the end paragraph of section 4.3 we add the following words to describe the motivation of calculating the mean of the potential energy in this altitude interval:

Author's changes in manuscript: "As the seasonal variations of  $E_p$  show semi-annual oscillation dominated features with 15 the approximated phases in the altitude range 87-97 km, the mean  $E_p$  in the range 87-97 km and the corresponding harmonic fit shown in Fig. 5b represents the behaviour of  $E_p$  well".

**Comments from referees:**

1. 5/13: Is the inversion also visible in the raw data or is it a result of the fit? If it is real, it should be discussed in more detail because it may strongly affect the propagation of GW. If it is not discussed here, some reference should be made.

20 Author's response: Yes, the inversion is visible in the raw data shown in Fig. 2 of the revised version. We have discussed the temperature inversion layer and its effects on the propagation of GW in section 5.1 and 5.2, respectively. Details is as following:

**Author's changes in manuscript:**

[revised manuscript text omitted]

**Comments from referees:**

15 1. 5/15: It remains open from this Figure, whether the mesopause could also be above 100 km in Sep-Dec.Author's response: Yes, it does.

**Comments from referees:**

5/20 and Fig. 2: It would be helpful to have contour lines to assess the similarity between the data sets.
 Author's response: The contour lines are overplotted in Fig.3a of the revised version.

**20 Comments from referees:**

1. 5/28: This is essentially expected if N2 is calculated from the mean temperature data set. See above. **Author's response**: Yes, it is not a finding but only an effect.

**Comments from referees:**

6/1-2: Please explain why it is worth to note this. My impression is that any kind of instability would be eliminated by
 building the temperature composite and applying an AO/SAO fit. Is the mesopause region more stable above Arecibo than somewhere else? If so, please explain and provide a reference.

Author's response: It is not worthy to note, consequently, this sentence is omitted.

1. 6/4-5: Consider plotting the phase of the SAO shifted by 180 d between 96 and 99 km (-80 will be +100). It may look nicer. Is the variability of the AO phase really unexpected (being the derivative of another property)? How does the variable phase affect the conclusions of the paper?

5 Author's response: this figure and the related Fig. 4c have been updated to make the shift of phase look nicer and the description has been updated as the following:

Author's changes in manuscript: "we allow negative amplitudes to make the phase oscillation of the two components look in order as shown in Fig. 4d. There are several turning points at altitudes  $\sim$ 87 km,  $\sim$  92km,  $\sim$ 94 km,  $\sim$ 96 km in the profiles of AO and SAO phases. The last three of these altitudes correspond to the altitudes of TILs. This agrees with the fact that the TILs locate at different altitude and N2 becomes large just below TIL (increased dynamical stability)."

**Comments from referees:**

10

1. 6/8/9: Do I understand correctly that the fitted T and N2 are used as the mean value to calculate T'? This would be a significant difference to Gardner&Liu. Furthermore, using the fitted time series will affect the seasonal variation of the derived GW activity. Please clarify.

15 **Author's response**: No, the nightly mean temperature is used to calculate T'. I just wanted to write a sentence connecting the preceding and the following, unfortunately, it is not correctly written. This sentence is omitted.

**Comments from referees:**

1. 7/9-10: I actually do not understand this sentence. Temperature enhancement from dissipating GW? Secondary waves? How relates a strength in temperature variation to a wind velocity?

20 Author's response: I mean that the smaller of the zonal mean zonal wind amplitude, the smaller the GW amplitude. But I can't discuss it further, it is omitted.

**Comments from referees:**

1. 7/13: How much of the increase above 97 km is due to the increasing uncertainty at the edge of the K layer or by a (potential) deviation of the fit data from the original data?

25 Author's response: the quantitative analysis is not given now.

**Comments from referees:**

1. 8/1: I do not agree that the phases of T and N2 are "random", even if they are less constant with altitude. A random phase would imply that AO/SAO are not relevant oscillations in this altitude range.

Author's response: the conclusion we drawn is subjective, the "are random" is updated to "vary" in this sentence.

1. 8/5-6: Please be more specific: Which processes do you expect that relate the GW activity to the seasonal wind variation in your data and what evidence you have?

Author's response: Thank you for this constructive comment which leads to the thorough rephrase of the discussion section and the conclusion section.

**Comments from referees:**

5

1. 8/9-10: This statement is rather vague. Please be more precise.

Author's response: We have rephrase this paragraph which is as the following:

Author's changes in manuscript: "The seasonal variations of GW potential energy are dominated by the combination of annual and semiannual oscillations at most altitudes. The maxima occur in spring and autumn and the minima occur during solstices. The observed GW potential energy is compared to the wind field as published by Garcia et al. (1997) and Smith (2002), There is indeed a pronounced altitudinal and temporal correlation between them. This suggests that the seasonal variation of GW activity should be determined mainly by the local wind field through the influence of critical level filtering of GW by the background wind."

**15 Comments from referees:**

Figure 2-4: Please provide error bars for fit amplitudes and phases.

Author's response: Figures 2-4 (figures 3-5 in the revised manuscript) have been replotted to draw the error bars for fitted amplitudes and phases according to this comment. The annual mean profiles of  $N^2$  and  $E_p$  are omitted in Fig. 4c and 5c in order not to mess the picture.

**20 Comments from referees:**

Figure 4: I recommend plotting the GWPED per volume. For linear propagation this should be conserved. The strong increase of GWPED close to 100 km would be less pronounced.

**Author's response**: Thank you for this suggestion. We have plotted the seasonal mean of GWPE per volume in Fig.6 of the revised manuscript and discussed the damp of GWPE in section 5.2.

**25 Comments from referees:**

Technical comments / Typos:

1. 3/19: "exhibits" should read "exhibited"

Author's response: thank you for pointing out this wrong spelling.

4/8: "evenly" should read "even"
 Author's response: It is corrected.

**Comments from referees:**

5 l. 5/2: "is" should read "are"Author's response: It is corrected.

**Comments from referees:**

l. 6/14: It is Fig. 4a

Author's response: It has been updated as Fig. 5a.

17

**Response to Editor Hibbins**
Before preparing and submitting a revised manuscript would the authors please provide a response to referee#1's specific comments relating to figures 2a and4a, and referee #2's first general comment. For clarity, these comments are reproduced below:

**10 **Referee #1**

Figure 2a - can you plot the MIL you refer to on the text on the figure

Author's response: This comment has been replied in the renewed response to the comments of referee #1.

**Comments from referees:**

Figure 4a – it might be easier to compare with other sites/lidar gw studies if you plot the lognormal of the GW PE.

15 Author's response: This comment has been replied in the renewed response to the comments of referee #1.

**Comments from referees:**

**Referee #2**

The authors describe in the Discussion a relation between the wind field as published by Garcia et al. (1997) and Smith (2012) and the observed variation of GWPED. While there is indeed a pronounced altitudinal and temporal correlation, the

20 paper lacks a description of the mechanism that relates the GW activity and zonal wind velocity. All statements are true, but remain vague and unspecific. The interpretation seems to imply pure zonal propagation of the waves, but the lidar data contain waves of all directions. Is the westerly wind between 60 and 70 km taken into account that may filter a lot of the eastward propagating GW?

Author's response: This comment has been replied in the renewed response to the comments of referee #2.

[revised manuscript text omitted]

---

## Author Response (AR2)

Two minor revisions:

**1. Comments from referees:**

1) In the abstract and throughout section 5.2 the use of the word "damp" is not quite correct. I would replace it with "reduction of GWPE" or the "GWPE reduces" for damps.

**Author's response**: Thank this referee for this comment, we have corrected the use of the word "damp" according to this comment.

**2. Comments from referees:**

2) There are still several places within the manuscript where the authors would benefit from a native English speaker looking over their paper to correct the grammar and word usage, e.g. pg 4, line 18 "estimate" should be "estimated"

**Author's response**: Thank this referee for the constructive comment. This revised version has been looked over by a native English speaker.

**Response to the Referee #2**

We would like to thank this referee for these constructive comments and suggestions to improve the quantity of the manuscript.

**1.  Comments from referees:**

The submitted paper is a revised version with substantial changes to the original paper. Most of the reviewer comments have been acknowledged. A Supplement is provided, with additional figures illustrating the evaluation method. Especially the discussion of the results is more elaborated, now. Nevertheless, there are still some points to clarify and I recommend the paper for revision. For example, some of the key arguments related to the connection of inversion layers and potential wave energy (new Fig. 6) need to be revised or further elaborated. Specific comments are given below.

**Author's response**:

The authors are grateful for the objective evaluation to the revised version. We would make further revision or elaboration according to the following comments.

**2.  Comments from referees:**

Specific                                                                                                         comments:

General: There are still some sentences with odd phrasing. Partly the content of the sentences is hard to comprehend because of odd word order or repetitions. Examples are p1l27-29, p2l5-7, p8l3-5, p10l19/20.

**Author's response**:

This comment helps to improve the phrasing of this manuscript. We have rephrased all these sentences listed in the comment.

**Author's changes in manuscript**:

p1l27-29, it was written "Their amplitudes grow rapidly with altitude under the condition of atmospheric density decreasing and begin to break or dissipate in the MLT where gravity wave influences have been shown to be strong by various observations". Now it is rephrased as "Their amplitudes grow rapidly with altitude under the condition of decreasing atmospheric density. GWs begin to break or dissipate in the Mesosphere and Lower Thermosphere (MLT) where their influences have been shown to be strong by various observations".

p2l5-7, it was written "The altitude ranges, where GWs interact with the background wind dissipating and deposing energy and momentum is of high interest to researchers.". Now it is rephrased as "The altitude ranges, where GWs interact with the background wind to dissipate and deposit energy and momentum to the background, is of high interest to researchers.".

p8l3-5, it was written "As the seasonal variations of $E_p$ show semi-annual oscillation dominated features with the approximated phases in the altitude range 87-97 km, the mean $E_p$ in the range 87-97 km and the corresponding harmonic fit shown in Fig. 5b represents the behaviour of $E_p$ well." Now it is rephrased as "In the altitude range 87-97 km, Fig. 5c shows that the amplitudes of both annual and semi-annual oscillations vary slightly , meanwhile, Fig. 5d shows that the phases of both annual and semi-annual oscillations vary also slightly. Therefore, the mean $E_p$ averaged in the altitude range 87-97 km and the corresponding harmonic fit shown in Fig. 5b can represent the seasonal behaviour of $E_p$ in this altitude range well.".

p10l19/20 it was written "The semiannual variation of GW intensity and mean zonal winds reported by Gavrilov et al. (2003) had been attributed to the dependence of GW generation and propagation on background wind and temperature by numerical simulations." Now it is rephrased as "Gavrilov et al. (2003) had attributed their observed semiannual variation of GW intensity to the dependence of GW generation and propagation on the background wind and temperature by numerical simulations".

**3.  Comments from referees:**

P2l17-20: Which part of this statement is only relevant for high latitudes and which for tropical latitudes? In order to keep the paper concise, you may limit the topic to tropical latitudes.

**Author's response:**

We have deleted the part relevant for high latitudes according to this comment.
**Author's changes in manuscript**:

The following words are deleted ", while it dominated in the range extending downwards to the stratopause in January or upwards to 110 km or higher in July at higher latitude".

**4.  Comments from referees:**

P2l21-23: If I understand correctly, you mix altitudes and seasons. This makes the sentence hard to comprehend. Please rephrase.

**Author's response:** This sentence has been rephrased according to this comment.

**Author's changes in manuscript**:

It was written: "The easterly winds were prevailing in equinoxes seasons near 80 km altitude. They then decreased with altitude from 80 km above and turned to increase above ~ 92 km, while the westerly winds prevailed in in the range 80-94 km in solstice seasons, they then turn to be easterly. the reversal is at about 95 km (Smith, 2012)."

It is now rephrased as: "the easterly winds were prevailing in equinoxes seasons in the 80-100 km altitude range with the minimal wind speed occurring near 92 km, while the westerly winds prevailed in solstice seasons in the 80-94 km altitude range with the maximal wind speed occurring near 88 km. The easterly winds turn to be prevailed above 95 km in solstice seasons (Smith, 2012)"

**5.   Comments from referees:**

P4l26-28: I do not understand how the large uncertainty at the edges justifies the limitation to 30 K RMS uncertainty. The uncertainty limits should be justified by geophysics, e.g. by the amplitude of the waves that shall be resolved. In this case, the GWs often have amplitudes smaller than 30 K, i.e. the data set should generally be limited to, e.g., 10 K RMSE like described later. Otherwise nights with higher uncertainty may compromise your analysis and, e.g., produce artificially high temperature fluctuations.

**Author's response:**

Thanks for this constructive suggestion. In the processing of the data, we first unify the raw data with resolutions of 0.45 km and 30 minutes into grids with resolutions of 0.9 km and 30 minutes. After that, the statistical uncertainty of the data improved significantly. Therefore, we choose 10 K RMSE in the data processing now.

**Author's changes in manuscript:**

"Step 1: for each night of observation, data points with photon noise errors larger than 30 K in temperature are discarded. The value of 30 K is set based on the fact that the root mean square errors (RMSE) due to photon noise often reach to about 40-50 K near the edges (~80 km on the bottom and ~105 km on the top) of the temperature profile." is rephrased as "Step 1: for the unified data with 0.9 km and 30 minutes in each night of observation, data points with photon noise errors larger than 10 K in temperature are discarded."

**6.   Comments from referees:**

P5l3/4: I understand that this method eliminates part of tidal temperature fluctuations. Please discuss the influence of remaining tidal variations with vertical wavelengths smaller than 50 km (and, e.g., 8 or 12 h period).

**Author's response:**

We discussed the influence of the remaining tidal variations for semidiurnal and terdiurnal tides at several places. For example, in the description of Fig. 2d, we write "The difference between Fig. 2c and Fig. 2d is pronounced. This indicates that step 4 is effective in eliminating the influences of the waves with longer vertical wave length. Furthermore, Guharay and Franke (2011) have given a not-weak semidiurnal tidal amplitude at the mesopause region through the meteor radar observations over a nearby site (20.75 °N, 156.43 °W). The amplitude increases with altitude and shows a clear SAO pattern with maxima during solstices. This kind of feature is not found in the variations of GW potential energy with altitude and

season in later sections here. However, the total effective of this method in removing the tidal is not easily to evaluated due to the lack of the knowledge on tidal components in this latitude firstly and the shorter and often intermitted measurement periods.";

**7. Comments from referees:**

5 P5l8-10: I do not understand how the temperature fluctuation composites are made without averaging the fluctuations, i.e. reducing the wave amplitude. Please explain.

**Author's response**:

I'm sorry that there is something wrong in describing the estimation of weekly mean temperature fluctuation. In the revision, firstly, the temperature deviations ($\Delta T$) obtained at each fixed vertical and time bins tare used to calculate the nightly mean
10 temperature perturbation profile with unified 0.9 km resolution. After that, the nightly mean temperature perturbation profiles within the same folded week are averaged to obtain the weekly mean temperature perturbation profile.

**8. Comments from referees:**

P5l10/11: From my understanding part of these inversions may in fact be remnants of GW. Meriwether and Gerrard (2004) provide a definition for MIL in terms of amplitude and persistence that should be used here, too. At least it should be
15 discussed why it is not used and how GW and TIL are separated.

**Author's response**:

Thank you to point out this point. It is not proper to talk about the TIL without removing the influences of GW. Therefore, this following sentence is deleted. "Temperature Inversion Layers (TILs) can be found in most months."
TILs are discussed after the seasonal temperature climatology is constructed by applying a harmonic fit to the weekly mean
20 temperature profiles. The definition for MIL by Meriwether and Gerrard (2004) is fulfilled then.

**9. Comments from referees:**

P5l12/13: Do I understand correctly that you first produce a time series of 7*24=168 h and then smooth this series by a 3h Hamming window? Is Jan 1 always day 1 of week 1?

**Author's response**:

25 Not yet. The process in computing the weekly composite night and the weekly mean profiles has been updated in the revision. It is as following: "The unified 0.9 km and 30 minutes resolution temperature profiles on each observational night in the same folded week are binned and averaged to the fixed same vertical and temporal grids to construct the weekly composite night of temperature of a year.

The weekly composite night data of $\overline{T}$ is first spatially and then temporally smoothed using Hamming windows with full widths at half maximum (FWHM) of 2.7 km and 3 hours, respectively. This method of computing the weekly composite night follows the approach used by Friedman and Chu (2007) and the references therein. The resulting weekly composite night of temperature usually covers most of the night from sunset to sunrise (not shown here, the resulting monthly composite night can be seen in Figure 3 of Friedman and Chu (2007), which is computed with a small part of the data set used here). It is a close representation of the mean state at the fixed time and altitude bins within an averaging week night. After that, the weekly composite nights are then averaged to derive the weekly mean profiles."

**10. Comments from referees:**

P6l1: N^2 is indeed highly variable with season and altitude. Unfortunately I did not find a discussion of the influence of this procedure on the results, i.e. the relation of the seasonal variations of temperature, N^2 and pot. energy.

**Author's response**:

Thank this referee for persisted in the discussion of this point. We really lack a discussion to this point in the early version. In section 4.2 of the revision, we add the following words: "However, it is noted that the seasonal variations of $N^2$ vary highly with altitude. This 87-97 km mean fitting curve cannot represent the whole features of $N^2$ in this altitude range."

**11. Comments from referees:**

P6l7-9: From Fig. 3a I read TIL at 95 km, 90 km, and 93 km, i.e. always 1 km lower than stated here. Additionally, some of the inversions are rather weak, i.e. potentially not robust with respect to the retrieval method and not compatible with Meriwether and Gerrard (2004).

**Author's response**:

Thanks for pointing out this deviation in labelling the TIL altitude (where the sign of temperature gradient change from plus to minus) in the figure. We have replotted the figures and checked to find that discrepancies occur after we reshaped the figure.

The monthly mean temperature profiles are plotted in the following figure. The profiles from left to right represent the results from January to December, respectively. each profile, except for the left most, is 10 K offset from the adjacent left one. The asterisks label the TIL altitudes. This result agrees well with that shown in Figures 4b and 5a in the revision and presents the inversions clearly. The inversions are rather weak in the winter months from November to January.

[Figure]

Fig. sup1 monthly mean profiles of the temperature over Arecibo. The asterisks represent the altitude where the sign of temperature gradient change from plus to minus.

**12. Comments from referees:**

P6l9/10: Is there an objective reason why this is the mesopause but cannot be a TIL?

**Author's response:**

This sentence has been updated to be as following: "The minimal temperature occurs near 98.5 km at most time except for the period from the second half of September to November where it is situated at ~ 96 km. This should be the mesopause according to the result that the mesopause is at the 95-100 km level at low-latitude obtained by SABER observations (Xu et al., 2007b)."

The following literature is added to the revision.

Xu, J., H.-L. Liu, W. Yuan, A. K. Smith, R. G. Roble, C. J. Mertens, J. M. Russell III, and M. G. Mlynczak (2007), Mesopause structure from Thermosphere, Ionosphere, Mesosphere, Energetics, and Dynamics (TIMED)/Sounding of the Atmosphere Using Broadband Emission Radiometry (SABER) observations, J. Geophys. Res., 112, D09102, doi:10.1029/2006JD007711.

**13. Comments from referees:**

P7l8-10: As already written in the first review, this is not a result, but a self-evident effect.

**Author's response:**

These sentences have been deleted. Meanwhile, more words are added to describe the features of $N^2$. They are as the following: "Figures 6 and 7a show that $N^2$ is highly variable with season and altitude. Take the summer months from June to

August for example, Fig. 7a shows that $N^2$ is quite low near the bottom and then increase quickly with altitude. It is obviously large between 87 and 92 km altitudes and decreases above. It is small in the 94-98 km range and turn to increase gradually near the top boundary. The seasonal variations are obviously different every 2-km-thick range. The features shown in Fig. 6b are more complicated than Fig. 7a. Assuming an isothermal atmosphere with the background temperature being 190 K, is estimated to be about $4.7\times 10^{-4}$ s$^{-2}$ which is represented largely by the orange color in figures 6b and 7a. Therefore, below about 96 km, the regions with red color match largely with the TILs. It shows clearly that the TILs occur at about 92-95 km altitude range in February and March while they occur at about 87-92 km altitude range through the summer months. A low value of GW potential energy is expected in the region with $N^2$ being large in case that other parameters kept."

**14.  Comments from referees:**

P7l15-18: I do not understand the reason for allowing negative amplitudes. Negative amplitude has no real physical meaning. Anyway, I do not see an advantage of plotting the phases "in order". There are indeed several phase turning points as expected from the "patchy" Fig. 4a. These patches of large N^2 agree with the TIL as expected (see above). But why do you expect the TIL (large N^2 regions) to have an AO/SAO and a specific altitude structure and how does Fig. 4d help to identify this. I am sorry, but in my eyes the phases of N^2 SAO/AO are fluctuating more or less randomly with altitude.

**Author's response**:

Thank you for this constructive comment. We have replotted this figure (now Fig. 7 in the revision) and keep the amplitude be positive. The description about the amplitude and phase has been updated as following:

"Figure 7c shows that the amplitudes of the 12-month and the 6-month oscillations are comparable throughout most of the altitude range of interest. They oscillate similarly and look like sinusoids with troughs occurring at altitudes ~87 km, ~ 92km, ~96 km, ~98 km. The phases of these two components illustrated in Fig. 7d also show quick transition at these altitudes."

**15.  Comments from referees:**

P8l1: The large RMSE of the phase for small amplitudes is not "odd", but exactly expected.

**Author's response**:

Thank you for telling this truth. This sentence is deleted in the revision.

**16.  Comments from referees:**

P8l25: How much of this increase is due to the fact that the uncertainty of the temperature measurement increases?

**Author's response**:

The relative error in the estimated $E_p$ could reach 30% or even larger due to the statistical uncertainty of temperature measurement at this altitude range. But the quantity influence of this uncertainty on the energy increase with altitude is hard to evaluated because the statistical uncertainty does not show a increase trend at 96-100 km altitude range as shown in Fig. 3b.

**17. Comments from referees:**

P9l2: In most of the months the mesopause is well above 100 km, but the increase of energy happens already >5 km below. Therefore the Offermann et al. mechanism cannot be used to explain the increase. Furthermore, Offermann et al. (2006) describe an exponential increase of wave amplitude, i.e. linear propagation. Here, the wave is gaining energy from some (non-linear) process that still needs to be identified.

**Author's response:**

Thank you for these two constructive comments. We have updated the text here.

In the last version, it is written as: "This result indicates that the GW reduces significantly dissipating or deposing energy below about the mesopause (or the wave-turbopause defined by Offermann et al. 2006), but it propagates upward almost freely after penetrating to the thermosphere."

In the revision, it is updated as: "This result suggests that a possible mechanism for the GW energy dissipation, i.e., the GW dissipates or deposes energy or momentum below about the mesopause (or the wave-turbopause defined by Offermann et al. 2006). This conjecture should be taken with caution. The relative error in the estimated $E_p$ could reach 30% or even larger due to the statistical uncertainty of temperature measurement at this altitude range. But the quantity influence of this uncertainty on the energy increase with altitude is hard to evaluated because the statistical uncertainty does not show a increase trend at 96-100 km altitude range as shown in Fig. 3b."

**18. Comments from referees:**

P9l14: I do not see a weaker damping of GW in winter below 94 km. But the tidal amplitude in Smith (2012) is reduced already above 80 km. From my understanding of the figures, the relation with the DW1 does not seem to be significant. Please provide further explanation or evidence.

**Author's response:**

Thank you for pointing out the flaw in this explanation. We delete this explanation in the revision.

**19.  Comments from referees:**

P9l21: (The sentence misses a predicate.) The gradient of the black curve starts to change at 97/98 km, not 96 km (or 95 km where the TIL is in fact). How relates the strong gradient change of the red curve at 97 km to the arguments presented here, if there is no TIL?

**Author's response:**

Thank you for pointing out these mistakes and providing a new thinking. We have corrected these mistakes and added the following sentence in the revision.

"From the GW energy point of view, the strong gradient change in the seasonal mean profiles of GW potential energy per unit volume should be impacted strongly by the horizontal wind field in this region."

**20.  Comments from referees:**

P9l27-32: I think there is a general flaw in the arguments. You relate the increase of potential energy in winter-spring-summer to the strong decrease of temperature with altitude. But for autumn you find the temperature increasing with altitude above 97 km, again resulting in an increase of potential energy. In fact, the increase cannot be explained solely by the temperature profile. As shown in equation 1 there are also contributions by N and T'. In linear theory (i.e. keeping energy constant), the variation of the amplitude of the fluctuations is a result of changes of N and mean-T, but a change of mean-T may not be able to increase the potential energy. From my point of view, an increase of potential energy (if real) can only be produced by a decrease in kinetic energy or by an additional "wave source" at this particular altitude. The latter in indirectly suggested later (p11l17) by the observational filter and Doppler shifts, but this argument needs to be further elaborated and proven.

**Author's response:**

Thank you for pointing out this general flaw and support the point about the mechanism of an additional wave source accounting for the energy increase. This mechanism will be discussed further in the latter subsection, therefore, this paragraph is deleted in the revision.

**21.  Comments from referees:**

P11l10/11: I do not understand how the zero-wind line at 96 km is responsible for the constant potential energy at this altitude. Waves typically propagate from below and therefore the energy at a specific altitude depends also on the conditions below, including the wave source.

**Author's response:**

The description for the fourth item is not proper, it is now updated as "Fourthly, the decrease of easterly winds with altitude near 85 km during equinoxes is accordance to the strong but decreasing GW $E_p$ with altitude in almost the same altitude range and seasons."

**22.  Comments from referees:**

P11l15-17: While Wright et al. (2016) had wind observations available, it remains speculative whether their arguments are valid her, too.

**Author's response:**

Thank you for mentioning that their arguments are still speculative. Therefore, it is not proper to cite it for explaining the results here. We delete this sentence in the revision.

**23.  Comments from referees:**

P11l24/25: What do you mean by "decrease rate is smaller than the lapse rate"? That there is a slower decrease (or even increase) of temperatures with altitude? Or that the temperature is decreasing similar to or even faster than the lapse rate, as many people find above MILs? Please rephrase to make this clearer.

**Author's response**:

This sentence is rephrased as "In altitude range of temperature decreasing, the decrease rate of background temperature with altitude is usually 1-2 K/km. Even right above an inversion layer, it is no larger than 4 K/km, which is extremely smaller than the lapse rate."

**24.  Comments from referees:**

Fig sup-1: The weekly averaged data look different to Fig. 2. Data gaps are at different times of the year. The strong gradient with altitude appearing at the time of the March-tickmark in Fig. 2 comes later in Fig. sup-1. Please check and explain, if real. Furthermore, I do not understand, why the lower part of the figure is processed differently to Fig. 2. You state that "The bottom panel is the harmonic fitted result $T(z, t)$ (estimated by using (3) in the revised manuscript)." If there is some further processing using the smoothed residuals, I wonder why.

**Author's response**:

Thank you for pointing out the deviation of the tick marker of the two panels. It has been corrected in the revision. Fig sup-1 has been added into the main text as Fig. 4 in the revision. To reproduce the variability of the weekly means, the lower panel is further processed using the smoothed residuals as you stated. The processing is as following:

"To keep the characteristics of higher order components in the seasonal climatology of each parameter, the fitted $\Psi(z,t)$ was subtracted from the raw weekly mean profiles and the residuals were smoothed using a Hamming window with a FWHM of 4 weeks. The smoothed residuals were then added back to the fitted $\Psi(z,t)$, the resulting seasonal climatology (hereafter SC) of each parameter is then illustrated together with weekly mean profiles in the results section.".

**25.  Comments from referees:**

Fig. sup-2 and sup-3: The fitted seasonal variations hardly reproduce the variability of the weekly means. In other words, SAO and AO are not the dominating features of these quantities, and the true dominating features may obscure the true SAO/AO.

**Author's response**:

These two figures are added to the main text and updated according the processing described in the response of the former comments.

**26.  Comments from referees:**

Fig sup-5: I strongly recommend to show this figure (together with a plot of the statistical uncertainty) in the main paper. It is necessary to understand the relevance of this study. E.g., it is very important to know that, in contrast to mid- and high latitudes, the perturbations (gravity waves) are dominated by small-scale variability and to a smaller extend by coherent wave structures (Fig. sup-5 d).

**Author's response**:

According to this comment, this figure has been inserted into the main body as Fig. 2. Also added is the temperature statistical uncertainty profiles and the corresponding uncertainty in the estimation of potential energy of gravity waves as Fig.3. To describe these two figures, the following texts have also been added.

"The data on 18 December 2003 are taken as an example of the data processing. Fig. 2a shows the temperature contours after step1. The data covers largely 82-103 km altitude range, the bottom edge increases to about 84 km in the second half of the night. Fig. 2b show the subtracted trends in time which indicates that the background temperature decreases at each altitude between 85-100 km range but increases near both the bottom and top edges. Fig. 2c is the corresponding temperature deviations. The temperature deviations after step 4 are shown in Fig. 2d. It shows that, in contrast to mid- and high latitudes (e.g. Baumgarten et al., 2018; Rauthe et al., 2008), the perturbations (gravity waves) are dominated by small-scale variability and to a smaller extend by coherent wave structures. The difference between Fig. 2c and Fig. 2d is pronounced. This indicates that step 4 is effective in eliminating the influences of the waves with longer vertical wave length. However, the total effective of this method in removing the tidal is not easily to evaluated due to the lack of the knowledge on tidal

components in this latitude firstly and the shorter and often intermitted measurement periods. Fig. 3a shows the individual profiles of the temperature fluctuations (cyan curves) for the night. The nightly mean temperature perturbations and the statistical uncertainties are also presented by black and red lines, respectively. The statistical uncertainties are usually less than 5 K below 95 km altitude, they increase a bit to ~ 6 K and keep it above. While the nightly mean temperature perturbations decrease from ~18 K at 85 km to ~6K at 89.5 km and then increase to and oscillate near 11 K above. The square of the ratio of temperature statistical uncertainty over nightly mean perturbation is plotted in Fig. 3b. It represents the relative error in estimating the potential energy due to statistical uncertainty in the temperature data. In this night, the relative errors are lower than 20% below 95 km and become larger above. The maximal relative error occurs just below 97 km with a value of 59%. The relative error is a bit larger than 30% near 100 km altitude but less than 10% near 85 km."

The sentence "**it is very important to know that, in contrast to mid- and high latitudes, the perturbations (gravity waves) are dominated by small-scale variability and to a smaller extend by coherent wave structures**" from this comment is adopted in describing the characteristics of the resulted temperature perturbations.

Minor/technical comments:

**27. Comments from referees:**

P4l9-11: Please make clear that these numbers are summed up over all years

**Author's response**: According to this comment, we have checked and updated table 1.

**28. Comments from referees:**

P5l20: "n-month oscillation"??

**Author's response**: "n" has been added in this sentence.

**29. Comments from referees:**

P6l32: "Equation (3)"

**Author's response**: "Equation" has been added before (3) in the revised version.

**30. Comments from referees:**

P9l18: "more rapid"

**Author's response**: We choose rapider here after look through the word "rapid" in the website: https://www.dictionary.com

**31.  Comments from referees:**

P10l15: delete "factor of"

**Author's response**: The words "factor of " have been deleted in the sentence: "The ratio between the maximum and the minimum is obviously larger…."

**32.  Comments from referees:**

Fig 3c: Please plot a dotted line for the SAO in the legend.

**Author's response**:

Thank you for pointing out this mistaken. It has been corrected in the revision.

**33.  Comments from referees:**

I did not mention various typos that should be easily identifiable with text processing software.

**Author's response:**

Thank this referee for this comment. We have checked up the whole article to minimize the typos.

[revised manuscript text omitted]

---

## Author Response (AR3)

**Response to the Referee #1**

**We would like to thank this referee for these constructive comments and suggestions to improve the quantity of the manuscript.**

**1. Comments from referees:**

The authors again improved the manuscript in this re-revised version, following the reviewer's comments. Most of the topics raised in the previous review have been properly answered. Some minor topics remain. I suggest acceptance of the paper after acknowledgment of these minor comments.

**Author's response**:

The authors are grateful for the objective evaluation to the revised version. We would make further revision or elaboration according to the following comments.

**2. Comments from referees:**

Equation 3: "365/" should read "365/7.

**Author's response**:

Thanks for pointing out this missing number. It is corrected in the revision.

**3. Comments from referees:**

P7l8: I think the deviations in autumn are not "minor". Depending on the evaluation method, some inversions either vanish or change in altitude by several kilometers. Considering the more or less zero temperature gradient, I suggest to ignore the whole period between September and December. (Maybe I missed the point, but why is the temperature maximum at 97 km not marked?)

**Author's response:**

Thanks this referee for pointing this improper word out and giving a constructive suggestion. We have rewritten this part according to this suggestion.

**Author's changes in manuscript**:

The sentence "There are minor deviations for the labelled altitude between these two figures." has been changed to "The deviations of TILs between these two figures are minor for most of the time. Significant differences occur mainly in the period between September and December. The inversion layers near 92 km in late September and near 89 km from late November to early December in Fig. 4b vanish in Fig. 5a, meanwhile, the inversion layer near 94 km during the first half of

September changes to ~ 92 km in Fig. 5a. It is noted that the temperatures vary slightly from 85 to 94 km altitude in most time from September to December. The inversion layers evaluated by finding the zero value of temperature gradient in this period are not confident enough in case of small perturbation existed, therefore, we ignore this period from September to December when discussing about the TILs. However, there is a common feature that temperature maximum occurs at ~ 97 in October and November in both figures 4b and 5a".

**4.   Comments from referees:**

P8l2-3: Please quantify "a bit".

**Author's response:**

Thanks this referee for pointing this imprecise description. We have rephrased this sentence after checking the data.

**Author's changes in manuscript**:

"the temperatures are a bit warmer in the winter months of December and January but are a bit colder near 100 km in March and April in this study" is now rephrased as: "the temperatures are about 10 K warmer in the winter months of December and January in this study".

**5.   Comments from referees:**

P8l5: I recommend rephrasing. For example "The second reason is that the harmonic fit model is applied to weekly averages here, while it was applied to monthly averages in Friedman and Chu (2007)."

**Author's response**:

Thanks for this constructive suggestion. This sentence has been rephrased just according to this comment.

**Author's changes in manuscript**:

"The second reason is the harmonic fit model is in term of week here, while it was in term of month in Friedman and Chu (2007)" has rephrased to "The second reason is that the harmonic fit model is applied to weekly averages here, while it was applied to monthly averages in Friedman and Chu (2007)".

**6.   Comments from referees:**

P10l16: "quantity" should read "quantitative".

**Author's response**:

"Quantity" has been replaced by "quantitative".

**7. Comments from referees:**

P10l30 to end of paragraph: The authors suggest that the TILs are the main driver for the GWEP increase, but this argument fails for autumn (red line), which is still not properly acknowledged. I agree that the horizontal wind might be responsible for this GWEP increase in autumn, but I think it is also responsible for the TIL. In other words, the TIL might not be the cause
5   for the GWEP increase, but the wind. Otherwise, the autumn profiles might be hard to explain. Another topic here: I am still missing some suggestion, why the GWEP increases instead of being constant with altitude as expected for freely propagating linear waves.

**Author's response**:

This comment contains two points, one is about the relationship between TILs and the GWEP variations, the other is about
10   the GWEP increase with altitude just below 100 km. We reply to them point to point.

For the first point:

Thanks this referee for this comment to make the relationship between TILs and GWEP variations clear. We agree with the comment that "the horizontal wind might be responsible for this GWEP increase in autumn, and it is also responsible for the TIL". We emphasized that "These close connections of the mesospheric TILs with the reduction of GW potential energy
15   provide strong support to the mechanism that the upper mesosphere TIL formed due to the interaction of GW with the upper mesospheric wind/diurnal tides through critical level effects. From other point of view, the strong gradient change in the seasonal mean profiles of GW potential energy per unit volume should be induced strongly by the horizontal wind field in this region." To illustrate the agreement between this comment and our view, we copy the relevant text in the following paragraph:

20   "The reduction of GW $E_p$ indicates the deposition of GW energy and momentum into the background atmosphere, which would lead to the increase of background temperature and/or even the occurrence of TIL. This drives us to investigate the relationship between the reduction of GW $E_p$ and the temperature structure in depth. We are excited to find that each profile of the GW potential energy per unit volume ((in $J.m^{-3}$) as shown in Fig. 10 shows a rapider reduction of energy at and below the TIL altitude of the corresponding season and turns to a much slower reduction and/or even conservation of energy
25   above. For examples, the behaviours of the green curve (profile for winter) around 94 km altitude (the altitude of TILs in winter), the blue curve (profile for summer) around 91 km altitude (the altitude of TILs in summer). The black curve (profile for spring) around 97/98 km altitude (~1 km above the altitude of TILs in spring). These close connections of the mesospheric TILs with the reduction of GW potential energy provide strong support to the mechanism that the upper mesosphere TIL formed due to the interaction of GW with the upper mesospheric wind/diurnal tides through critical level

effects. From other point of view, the strong gradient change in the seasonal mean profiles of GW potential energy per unit volume should be induced strongly by the horizontal wind field in this region."

For the second point:

Thanks this referee for this valuable suggestion, "why the GWEP increases instead of being constant with altitude as expected for freely propagating linear waves". Possible reasons for the increase of GWEP includes GWs generated locally and GWs propagated meridionally from middle or high latitudes. The increases of GWEP in Spring and Autumn above 95 km altitude associate with the increased zonal wind shears as shown in figure 1 of Li et al. (2012), which suggests that new GWs be generated by the dynamical instability.

This discussion has been added in the end of section 5.3 as the following: "It is noted that the GWEP increases instead of being constant with altitude as expected for freely propagating linear waves above 98 km in Spring and above 96 km in Autumn as shown in Fig. 10 and Fig. 9a. We notice that the corresponding regions of increasing GWEP enhancement with altitude in Fig. 9a associate well with the regions of increasing zonal wind shears in Figures 1a and 1b in Li et al. (2012), which suggests that conversion of energy contained in the mean wind into the energy of perturbation due to the increasing dynamical instability should be a most possible reason for the enhancement of GWEP with altitude."

An additional item has been included in the conclusion section as the following: "4. The GWEP increases near 100 km latitude in equinox seasons instead of being constant with altitude as expected for freely propagating linear waves. This most possibly caused by the conversion of energy contained in the mean wind into the energy of perturbations due to dynamical instability."

**8. Comments from referees:**

P13l6: I suggest writing "September to December" instead of "October to November", as explained above.

**Author's response**:

This replacement has been done according to this comment.

**9. Comments from referees:**

P13l8: I assume that this is at least partly caused by the averaging. For each single night, the gradient might be closer to the lapse rate.

**Author's response**:

This comment suggests that the conclusion might be the result of averaging, therefore, we omit the following sentence "In altitude range of temperature decreasing, the decrease rate of background temperature with altitude is usually 1-2 K/km.

Even at altitudes right above an inversion layer, the decrease rate is no larger than 4 K/km, which is extremely smaller than the lapse rate."

**10. Comments from referees:**

Supplement: The supplementary figures now partly repeat the main figures. Please reduce the Supplement accordingly.

5   **Author's response**:

We have reduced the Supplement accordingly.

[revised manuscript text omitted]

---

## Author Response (AR4)

**Response to Co-editor Robert Hibbins**

**We would like to thank Co-editor Robert Hibbins for these constructive comments to improve the quantity of the manuscript and his valuable work to push this article to be published.**

**1. Comments:**

The authors have gone a long way to address the issues raised by the reviewers in this iteration of the manuscript. I would be grateful if they could clarify the following points in a minor revision to the manuscript before publication.

**Author's response**:

The authors are grateful for this kind and constructive comments. The first author is sorry for the inexact responses to the issues raised by the reviewers in the last iteration of the manuscript.

**2. Comments:**

P12 L16 "The inversion layers evaluated by finding the zero value of temperature gradient in this period are not confident enough in case of small perturbation existed, therefore, we ignore this period from September to December when discussing about the TILs. However, there is a common feature that temperature maximum occurs at ~ 97 in October and November in both figures 4b and 5a." The language is poorly phrased. Perhaps the authors mean something like: "The inversion layers, evaluated by finding the zero value of temperature gradient, are not statistically significant in this period. Therefore, we ignore the period from September to December when discussing the TILs. However, there is a common feature of a temperature maximum at ~ 97 km in October and November in both figures 4b and 5a." Please clarify and tidy up the English used in this important explanation.

**Author's response**:

Thank Robert for rephrasing this explanation. Just as this comment, we mean "The inversion layers, evaluated by finding the zero value of temperature gradient, are not statistically significant in this period. Therefore, we ignore the period from September to December when discussing the TILs. However, there is a common feature of a temperature maximum at ~ 97 km in October and November in both figures 4b and 5a."

**Author's changes in manuscript**:

P7 L10 in the manuscript uploaded in this iteration (P12 L16 in Author's response of the last iteration): The sentence "The inversion layers evaluated by finding the zero value of temperature gradient in this period are not confident enough in case of small perturbation existed, therefore, we ignore this period from September to December when discussing about the TILs. However, there is a common feature that temperature maximum occurs at ~ 97 in October and November in both figures 4b

and 5a" has been replaced by "The inversion layers, evaluated by finding the zero value of temperature gradient, are not statistically significant in this period. Therefore, we ignore the period from September to December when discussing the TILs. However, there is a common feature of a temperature maximum at ~ 97 km in October and November in both figures 4b and 5a."

5 **3.   Comments from referees:**

GWPE and GWEP are now used interchangeably in the manuscript. Be consistent (I recommend use GWPE throughout).

**Author's response:**

Thank Robert for pointing this mistake. GWEP has been replaced by GWPE in the revised manuscript.

**4.   Comment:**

10   P18 L11 "associated well with the regions" - do the authors mean "are clearly associated with regions"?

**Author's response**:

Yes, we mean "are clearly associated with regions". This sentence has been rephrased just according to this comment.

**5.   Comment:**

P18 L13 "should be a most possible reason" - it is not clear what this means. Please clarify in unambiguous language.

15 **Author's response**:

"should be a most possible reason" is changed by "should be an important mechanism" in the revised manuscript.

[revised manuscript text omitted]